# Retrieval of cloud fraction and optical thickness of liquid water clouds over ocean from multi-angle polarization observations

Claudia Emde[1,2], Veronika Pörtge[1], Mihail Manev[1], and Bernhard Mayer[1]

[1]Meteorologisches Institut, Ludwig-Maximilians-Universität München, Germany
[2]Institut für Physik der Atmosphäre, Deutsches Zentrum für Luft- und Raumfahrt (DLR), Germany

**Correspondence:** Claudia Emde (claudia.emde@lmu.de)

**Abstract.**

We introduce a novel method to retrieve the cloud fraction and the optical thickness of liquid clouds over a water surface based on polarimetry. The approach is well-suited for satellite observations providing multi-angle polarization measurements, in particular the Hyper-Angular Rainbow Polarimeter #2 (HARP2). Unlike commonly used methods to derive the cloud fraction our method does not depend on the spatial resolution of the observations, and it does not require any threshold values for cloud detection. Based on radiative transfer simulations we show that the cloud fraction and the cloud optical thickness can be derived from measurements at two viewing angles: one within the cloudbow and a second in the sun-glint region. In the cloudbow, the degree of polarization depends mainly on the cloud optical thickness. Conversely, for a viewing direction in the sun-glint region, the degree of polarization depends on the clear fraction of the pixel, because here the radiation scattered by cloud droplets is almost unpolarized whereas radiation reflected by the surface is highly polarized. Utilizing these dependencies, we developed a retrieval using a simple lookup-table approach. Based on sensitivity studies, we show that prior information about wind speed and aerosol optical thickness improves the accuracy of the cloud fraction retrieval. Prior information about the cloud droplet size distribution can reduce the uncertainty of the cloud optical thickness retrieval. The prior information should be obtained by combining our method with already existing aerosol and cloud retrieval algorithms. We performed 3D radiative transfer simulations and found that the cloud optical thickness is generally underestimated due to neglecting 3D scattering effects. The cloud fraction is overestimated in cloud shadows and underestimated in the in-scattering regions.

As a demonstration, we apply the methodology to airborne observations from polarization cameras of the Munich Aerosol Cloud Scanner (specMACS) instrument. The high spatial resolution data (10-20 m) has been averaged to a spatial resolution of approximately 2.5 km to mimic satellite observations. From the average linear polarization at scattering angles of 140° and 110° we derive continuous cloud fraction values and the corresponding cloud optical thicknesses. The comparison for cases including low, medium, and high cloud fractions shows that the retrieval, using only the reflected polarized radiances at two scattering angles, provides accurate estimates of the cloud fraction for observations with coarse spatial resolution.

# 1 Introduction

Clouds influence local weather conditions as well as the Earth's climate system. They affect the energy balance and play a large role in the planet's long-term climate. According to the latest IPCC assessment report AR6 (Intergovernmental Panel On Climate Change, 2023), there have been major advances in the understanding of cloud processes over the last decade that have decreased the uncertainty range for the cloud feedback by about 50% compared to AR5 (Intergovernmental Panel On Climate Change, 2014) but nevertheless, clouds remain the largest contribution to the overall uncertainty in climate feedbacks.

In order to further improve the representation of clouds in climate models observations are required for model validation. The Cloud Assessment Group of the GEWEX (Global Energy and Water Exchanges) program gathers cloud products derived from active and passive satellite observations in the solar and thermal spectral regions (Stubenrauch et al., 2024) and provides a database of publicly available, global cloud products at a spatial resolution of 1° latitude × 1° longitude. They also provide recommendations on how satellite-retrieved cloud properties may be used in climate studies and climate model evaluation. One important parameter is the global cloud fraction, for which they obtain an average of $0.66\pm0.04$ from eleven participating datasets. The standard method to derive cloud fraction from space is to calculate the fraction of image pixels that contain some clouds. E.g., for MODIS, which has a relatively high spatial resolution of about $1\times1$ km$^2$, the cloud fraction is retrieved on a spatial resolution of $5\times5$ km$^2$ by computing the ratio of pixels where clouds have been detected and the clear pixels for this larger region including 25 individual MODIS pixels (Platnick et al., 2017). The determination of cloud fraction from satellite measurements is problematic for various reasons: First, there is no quantitative cloud definition, e.g. a lower liquid water content limit. Secondly, most cloud detection methods rely on thresholds, which depend on instrument and algorithm performance as well as on cloud assumptions and on the background. Thirdly, the definition of cloud fraction based on the fraction of cloudy pixels in an image depends on the viewing direction and strongly on the spatial resolution of the observations (e.g. Wielicki and Parker, 1992).

Di Girolamo and Davies (1997) developed a pattern recognition approach to correct for the spatial distribution error. Dutta et al. (2020) applied this method to correct for the spatial resolution error in Multi-angle Imaging SpectroRadiometer (MISR) observations and found cloud fraction reductions of more than 0.4 in regions dominated by shallow cumulus clouds. They validated the resolution-corrected cloud product by comparison to Advanced Spaceborne Thermal Emission and Reflection Radiometer (ASTER) observations of 15 m resolution and show that the 50°N to 50°S cloud fraction, which is in accordance to the GEWEX assessment of about 0.65 in the uncorrected MISR cloud product, is reduced to 0.47 in the resolution corrected cloud product.

An alternative resolution-independent cloud fraction retrieval approach is implemented in the aerosol retrieval algorithm based on optimal estimation by Hasekamp (2010), which provides, in addition to aerosol optical properties, the cloud fraction from POLDER measurements with a spatial resolution of about $7\times6$ km$^2$. In a validation study for partially cloudy scenes based on simulated data, Stap et al. (2016a, b) show that the retrieved cloud fraction correlates well with the cloud fraction used as input for radiative transfer simulations to produce the simulated data. Similarly, Van Diedenhoven et al. (2007) developed a retrieval method for cloud parameters from satellite-based reflectance measurements (Global Ozone Monitoring Experiment

and Scanning Imaging Absorption Spectrometer for Atmospheric Chartography) in the ultraviolet and the oxygen A-band. Based on an optimal estimation approach they derive cloud fraction, cloud optical thickness, and cloud top pressure. The oxygen A-band contains information about cloud optical thickness and cloud top pressure. The UV spectral region is also sensitive to cloud fraction, since the Rayleigh scattering contribution to the reflectance increases with the clear fraction of the pixel. Recently, another cloud fraction algorithm for PARASOL data based on a neural network has been developed. The training data includes variations in aerosol and surface properties as well as ice and liquid clouds and it therefore can be applied globally (Yuan et al., 2024). Note that these algorithms do not rely on thresholds for cloud detection because the cloud fraction is not defined by a fraction of cloudy pixel but solely derived from the observed reflected radiation and it is therefore a resolution-independent continuous quantity.

In this study, we propose a similar method to estimate the cloud fraction of liquid water clouds over ocean. It makes use of the angular dependence of the degree of linear polarization of reflected radiation. For clouds, it becomes large in the cloudbow region, whereas for clear sky above ocean, it becomes large in the sun-glint region. Therefore, we propose to measure the degree of linear polarization at two angles, one in the cloudbow region to retrieve the vertical cloud optical thickness of the cloudy part and another angle in the sun-glint region to retrieve cloud fraction.

The global cloud cover cannot be obtained using our method because it is developed only for liquid clouds over ocean. Since ice clouds do not produce a cloudbow, the methodology cannot directly be applied to determine the cloud fraction and the optical thickness of ice clouds. However, it would be possible to replace the degree of linear polarization in the cloudbow region by an intensity observation at the same angle to retrieve the ice cloud optical thickness. For retrievals over land, polarization due to surface reflection is too small to be used for a cloud fraction retrieval. Therefore an additional method needs to be developed which could use the strong polarization caused by Rayleigh scattering between the clouds to obtain information about the cloud fraction. This method should use shorter wavelengths which are mostly insensitive to surface properties and for which the Rayleigh scattering contribution is much higher.

The resolution-independent cloud fraction retrieval method could be a valuable addition to operational cloud retrieval algorithms for upcoming satellite instruments providing multi-angle polarization observations. The NASA Plankton, Aerosol, Cloud, ocean Ecosystem (PACE) mission (https://pace.oceansciences.org) which has successfully been launched on 8th of February 2024 includes two polarimeters, the Spectro-polarimeter for Planetary Exploration (SPEXone) and the Hyper-Angular Research Polarimeter #2 (HARP2) (Remer et al., 2019). SPEXone provides hyper-spectral polarized radiances in the spectral range from 385–770 nm in five viewing directions for a narrow swath of 100 km at nadir. The spatial resolution is about 5 km and global coverage is obtained after approximately 30 days. HARP2 is a hyper-angular instrument, with four spectral bands between 440 nm and 870 nm and observes a wide swath of 1555 km at nadir. For the 669 nm band HARP2 includes 60 viewing angles spaced over 114°. The spatial resolution is approximately the same as SPEXone. The Multi-Viewing Multi-Channel Multi-Polarisation Imaging (3MI) instrument is an optical radiometer dedicated to aerosol and cloud characterization (Fougnie et al., 2018). It is one of the missions of the EUMETSAT Polar System Second Generation (EPS-SG) program planned to be launched in 2025. It will provide a multi-spectral (from 410 to 2130 nm), multi-polarisation (-60°, 0°, and +60°), and multi-

angular (14 views) image of the outgoing radiance at the top of the atmosphere. The spatial resolution is 4 km at nadir and the swath width is 2200 km.

Since cloud structures change rapidly, observations at the two suggested scattering angles should ideally be taken nearly simultaneously. HARP2 is designed to provide such observations, so it is ideally suited for our proposed method. For SPEXone and 3MI, further investigations on the collocation of the observations would be required.

The paper is structured as follows: In Section 2 the setup of the radiative transfer model for 1D and 3D simulations is described. In Section 3, the dependence of the degree of polarization on cloud optical thickness and cloud fraction is investigated and retrieval lookup tables for different atmosphere and surface conditions are constructed. In Section 4, 3D scattering effects on the cloud fraction and cloud optical thickness retrieval are investigated. In Section 5, we apply the retrieval method to airborne observations of the specMACS instrument. The final Section 6 includes a brief summary, discusses limitations and provides an outlook on future work.

## 2 Methodology

For all simulations we used the radiative transfer model MYSTIC (Monte carlo code for the phYsically correct Tracing of photons In Cloudy atmospheres, Mayer, 2009; Emde et al., 2010) implemented in the libRadtran package (Mayer and Kylling, 2005; Emde et al., 2016). MYSTIC is a comprehensive vector radiative transfer model that can be run in 1D or 3D, plane-parallel or spherical geometry. It has been extensively validated in various model intercomparison studies (e.g., Emde et al., 2015, 2018).

The general setup for all simulations is as follows: we take the US-standard atmosphere from Anderson et al. (1986) to set up the profiles of pressure, temperature, and trace gas concentrations. As incoming solar irradiance we use the extraterrestrial spectrum by Kurucz and Bell (1995). We enable the polarisation mode (Emde et al., 2010) to compute the complete Stokes vector, and the variance reduction methods (Buras and Mayer, 2011) for accurate simulations including cloud scattering. We perform monochromatic simulations at 667 nm, approximately the center wavelength of the HARP2 instrument where 60 scattering angles are observed simultaneously. The solar zenith angle is set to 50°. The simulations are performed for viewing angles from -60° to +60° in steps of 1° in the solar principal plane, thus we obtain the Stokes vector for scattering angles $\Theta$ from 70° to 180°. The scattering angle is defined as the angle between the sun position vector (incident direction) and viewing direction vector (i.e., the sun is behind the observer at 180°). As we will show later (see Fig. 4 (g)), the retrieval performs best for observations in the solar principal plane including the maximum of the sun-glint.

The components of Stokes vector $\mathbf{I}$ are defined as time averages of linear combinations of the electromagnetic field vector (e.g., Chandrasekhar, 1950; Hansen and Travis, 1974):

$$I = \left\langle E_\parallel E_\parallel^* + E_\perp E_\perp^* \right\rangle, \tag{1}$$

$$Q = \left\langle E_\parallel E_\parallel^* - E_\perp E_\perp^* \right\rangle, \tag{2}$$

$$U = \left\langle E_\parallel E_\perp^* + E_\perp E_\parallel^* \right\rangle, \tag{3}$$

$$V = i\left\langle E_\parallel E_\perp^* - E_\perp E_\parallel^* \right\rangle. \tag{4}$$

Here, $E_\parallel$ and $E_\perp$ are the components of the electric field vector parallel and perpendicular to the reference plane, respectively. $I$ is the intensity of the radiation, $Q$ and $U$ give the state of linear polarization and $V$ the circular polarization. The unit of the Stokes components is $W/(m^2 \, nm \, sr)$. We will neglect $V$ in the following because circular polarization is several orders of magnitude smaller than linear polarization (e.g., Emde et al., 2018). In the solar principal plane, $U$ is exactly 0 for plane-parallel geometry by definition. Therefore the signed degree of linear polarization is given by

$$P = Q/I \tag{5}$$

A negative (positive) $P$ means that the radiation is predominantly polarized perpendicular (parallel) to the scattering plane. Note that $P$ is a dimensionless quantity which can be measured without absolute calibration. For 3D geometry, $U$ can be non-zero and we use the following equation to calculate the signed degree of linear polarization:

$$P = -\frac{\sqrt{Q^2 + U^2}}{I} \tag{6}$$

The negative sign indicates that the radiation is predominantly polarized in the direction perpendicular to the scattering plane, i.e. $P$ has the same sign as $Q/I$ in the solar principal plane.

In the following, we call the directional dependence of radiance on the scattering angle $\mathbf{I}(\Theta)$ the "phase curve" and the directional dependence of the degree of polarization $P(\Theta)$ the "polarized phase curve".

The polarized bidirectional reflectance distribution matrix of the ocean surface is modeled using the reflectance matrix based on the Fresnel equations convolved with a Gaussian kernel to account for the ocean waves (Mishchenko and Travis, 1997; Tsang et al., 1985; Cox and Munk, 1954a, b). Water surface reflection causes very strong polarization in the sun-glint. The wind speed which determines the spread of the sun-glint region is set to 5 m/s if not specified otherwise. In this study, land surfaces are assumed to be Lambertian reflectors, i.e. reflected radiation is completely unpolarized.

## 2.1 Model setup for 1D radiative transfer simulations and independent pixel approximation

All 1D simulations are performed for a clear sky atmosphere and for a liquid cloud layer with a geometrical thickness of 1 km and a cloud top height of 3 km, respectively. The optical thickness of the cloud is varied between 1 and 50 (the values are set to 1,3,5,10,20,50). Cloud droplet sizes are gamma distributed with an effective radius of 10 μm and an effective variance of 0.1. Cloud optical properties were calculated using the Mie program included in libRadtran (Mie, 1908; Wiscombe, 1980). Note

that this simple setup is only used to illustrate the main sensitivities to cloud fraction and cloud optical thickness. Sensitivity studies with different model parameters are performed later in Sections 3.2 and 3.3.

In order to calculate the Stokes vector for a partially cloudy pixel we combine the clear and the cloudy simulations, which is commonly called independent pixel approximation (IPA):

$$\mathbf{I} = (1 - c) \cdot \mathbf{I}_{\text{clear}} + c \cdot \mathbf{I}_{\text{cloudy}} \tag{7}$$

Here $c$ is the cloud fraction of the pixel, $\mathbf{I}_{\text{clear}}$ is the Stokes vector simulated for clear sky, and $\mathbf{I}_{\text{cloudy}}$ is the Stokes vector simulated for the cloudy sky. The degree of linear polarization of a partially cloudy pixel in the principal plane is

$$P = -\frac{(1-c)Q_{\text{clear}} + cQ_{\text{cloudy}}}{(1-c)I_{\text{clear}} + cI_{\text{cloudy}}} \tag{8}$$

Note that it cannot be calculated as the linear combination of the individual degrees of polarization.

## 2.2 Model setup for 3D radiative transfer simulations

### 2.2.1 2D cloud scene with sharp cloud edge

In order to quantify the impact of 3D cloud scattering systematically we define a scene, where half of the domain is cloud covered and the other half is clear sky. In this case, we can distinguish the two basic 3D effects: in-scattering, when the cloud side is illuminated, and shadowing, when the cloud casts a shadow on the surface. Such a sharp cloud edge is of course an extreme case for which we may expect the most significant 3D effects.

The 2D cloud is included in the same background atmosphere that is used for the 1D simulations (Section 2.1). The cloud height is between 1 and 2 km. As before the effective radius of the cloud droplets is set to 10 μm and the vertical optical thickness is varied between 1 and 50. The wind speed is set to 5 m/s and the simulations are performed without aerosols. The domain size was set to 100 km with the cloud edge in the center. We simulate polarized observations of the step cloud at the top of the atmosphere at a spatial resolution of 500 m in $x$-direction.

### 2.2.2 Randomly distributed box clouds

The second scenario should resemble shallow cumulus cloud fields with different cloud fractions. We define 5×5 pixels with a spatial resolution of $500 \times 500$ m$^2$ and randomly fill the pixels with clouds, this way we get cloud fields with cloud fractions of $\{1/25, 2/25, ...25/25\}$. The geometrical thickness of the clouds is 100 m, typical for shallow cumulus fields (e.g. Zhan et al., 2021). MYSTIC applies periodic boundary conditions, that means that the same clouds are repeated horizontally in $x$ and $y$ directions. The background atmosphere and cloud microphysical properties are the same as for the 2D cloud edge (Section 2.2.1). We include an ocean surface and as before the wind speed is set to 5 m/s. We sample polarized radiances at the top of the atmosphere at a spatial resolution of $500 \times 500$ m$^2$.

## 3 Setup of the retrieval

In this section we analyze the sensitivity of the polarized phase curve on cloud fraction and cloud optical thickness based on 1D radiative transfer simulations and the IPA approximations. In addition, we investigate the impact of wind speed, aerosols, and cloud altitude.

### 3.1 Phase curves for broken liquid clouds above an ocean surface

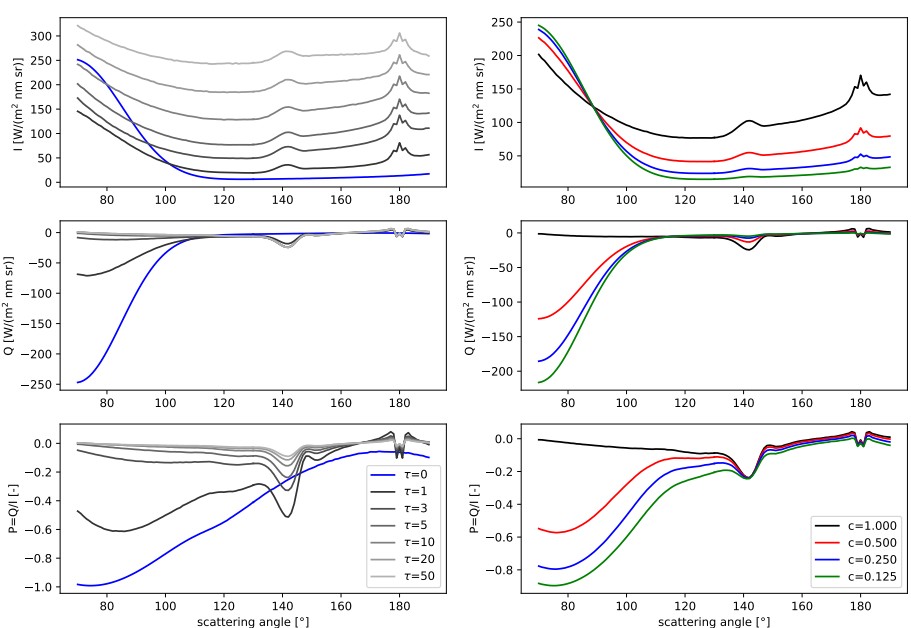

**Figure 1.** *Left:* Sensitivity of phase curve on cloud optical thickness $\tau$ at 667 nm. *Right:* Phase curve for fixed $\tau = 5$ for various cloud fractions. All simulations are for an ocean surface, a wind speed of 5 m/s and a solar zenith angle of 50°. Cloud droplets sizes are modeled using a gamma distribution with an effective radius of 10 μm and an effective variance of 0.1.

Results of the 1D simulations including an ocean surface and a cloud layer as defined in Section 2.1 are shown in the left panel of Figure 1. The blue line corresponds to the clear sky simulation. The top panel shows the total intensity $I$, the middle panel the linearly polarized intensity which is equal to the $Q$-component of the Stokes vector in the solar principal plane, and the bottom panel is the degree of linear polarization $Q/I$. The $U$-component of the Stokes vector is zero in the principal plane. The figures clearly show the broad sun-glint region at scattering angles around 100°. The degree of linear polarization of the sun-glint is close to 1 for clear sky (see blue line in lower left panel). The $Q$-component (middle left panel) is negative which means that the polarization direction is perpendicular to the scattering plane.

The grey lines in the left panels correspond to various cloud optical thicknesses from 1 (dark grey) to 50 (light grey). The intensity (unpolarized radiance $I$) increases with increasing cloud optical thickness since more radiation is reflected. The linearly polarized radiance $Q$ saturates relatively quickly around $\tau = 5$. All curves show two distinct features, the cloudbow at

scattering angles around 140° and the backscatter glory around 180°. In particular the cloudbow is highly polarized. Looking at the $Q$-results we find that the linear polarization predominantly emerges from surface reflection at scattering angles around 80° and mainly from cloud scattering at scattering angles around 140°.

We then calculate the Stokes vector for a fixed cloud optical thickness of 5 for cloud fractions between 1/8 and 1 (fully cloudy), the results are shown in the right panels of Fig. 1. The black line corresponds to full cloud cover ($c$=1) and is identical to the line for $\tau$=5 in the left panels. When we focus on the degree of linear polarization $Q/I$ we see that it does not depend on cloud fraction in the cloudbow region, whereas there is a strong dependence on cloud fraction in the glint region. The reason is that surface reflection causes strong polarization for angles smaller than about 110°. For partly cloudy pixels a part of the

surface is seen by the observer, and therefore, the degree of polarization is increased in glint directions compared to fully cloudy pixels. In the cloudbow region, only the cloud contributes to the degree of linear polarization, therefore the degree of linear polarization is not changed when a part of the surface becomes visible.

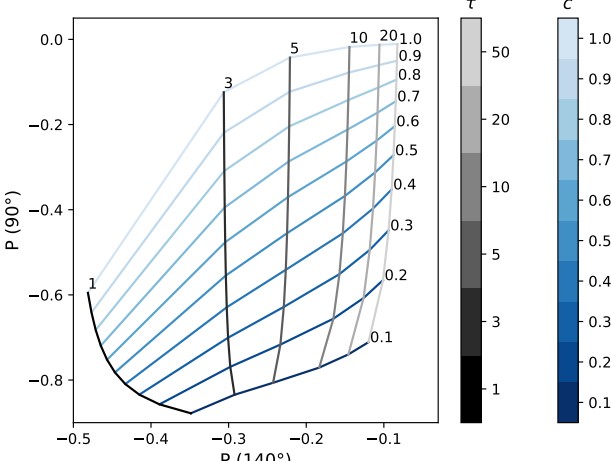

**Figure 2.** Lookup table for cloud cover and cloud optical thickness retrieval. The blue lines correspond to constant cloud fractions and the grey lines correspond to constant cloud optical thickness values.

    Using these dependencies allows us to generate a retrieval lookup table as illustrated in Figure 2. Here, we plotted the degree of polarization at $\Theta$=140° ($P(140°)$) on the x-axis and the degree of polarization at $\Theta$=90° ($P(90°)$) on the y-axis. The blue

lines correspond to constant cloud fraction values, i.e. the upper light blue line is for a cloud fraction of 1.0 (fully cloudy) and the lower dark blue line is for a cloud fraction of 0.1. The grey lines correspond to constant cloud optical thickness values between 1 (dark grey) and 50 (light grey). The lookup-table plot illustrates, that cloud fraction and cloud optical thickness can be retrieved from the degree of linear polarization observed at the two viewing angles since the blue and grey lines separate nicely. Note that, since the cloud fraction is derived from the observation in the sun-glint, the retrieved cloud fraction

corresponds to the cloud fraction of the pixel observing the glint.

## 3.2 Retrieval lookup tables for various scenarios

Fig. 3 shows polarized phase curves and the corresponding lookup tables for various scenarios. The upper row (a) corresponds to the scenario presented in the previous section, with an ocean surface and a wind speed of 5 m/s.

To investigate the influence of wind speed on the polarized phase curve, we conducted identical simulations with a wind speed of 10 m/s (scenario (b)). We find that the maximum of $|P|$ for partially cloudy pixels is decreased and also the slope of $|P|$ is smaller for scattering angles between 90° and 110°. The lookup table generated for higher wind speeds looks similar to that of the previous case and is just as good to retrieve optical thickness and cloud fraction.

The presence of aerosols also modifies the polarization state. Therefore, we repeated the simulations with additional aerosols corresponding to the mixture "maritime clean" as defined in the OPAC database (Hess et al., 1998; Emde et al., 2016) with an aerosol optical thickness set to 0.1 (scenario (c)). Compared to the results without aerosols, we find that the degree of polarization is slightly decreased due to increased multiple scattering, as expected. Furthermore, the polarized phase curves closely follow the same pattern as the scenario (a) and also the lookup table appears nearly identical.

Finally, we test whether the methodology would also work for land surfaces. We use a Lambertian surface as an approximation which depolarizes the reflected light completely. This is a realistic approximation as the largest polarized reflectances[1] observed by the PARASOL instrument over land are in the range between 0.02 and 0.04 (Maignan et al., 2009). Row (d) of Fig. 3 shows the simulations for a dark surface (albedo=0.0, scenario (d)). The blue line in the left figure shows the simulation for clear sky and we see the high degree of polarization around 90° scattering angle. When a cloud is added, the degree of polarization is smaller than for the corresponding cases over ocean, because the polarized reflectance from Rayleigh scattering is much smaller than from reflection in the sun-glint and the signal of the cloud dominates. The lookup-table plot indicates that, for small cloud optical thicknesses and low cloud fractions, the lines are distinct, but as the cloud optical thickness increases, the lines converge. This convergence may lead to less accurate retrieval results for the same measurement accuracy. If the surface albedo is nonzero, this situation becomes worse. Row (e) of Fig. 3 shows the results for a surface albedo of 0.2. In this case the surface depolarizes and the entire lookup table is compressed making a retrieval impossible.

## 3.3 Dependence on various parameters, e.g. wind speed, aerosol properties, cloud microphysics

To assess the robustness of the retrieval method we performed additional sensitivity studies which are presented in Fig. 4. The base case for all simulations is defined as follows: ocean surface, cloud layer at 2-3 km altitude with an optical thickness of $\tau_c$=5, cloud droplet effective radius 10 μm, wind speed 5 m/s, aerosol optical thickness $\tau_a$=0.1, solar zenith angle 50°. Starting from the base case, one of the parameters is varied whereas all other parameters are kept constant. We also include results for a scattering angle of 110°, not in the maximum of the sun-glint. The panels in the left column of Fig. 4 show the sensitivities for $P(140°)$, the middle ones correspond to $P(110°)$, and those in the right column are for $P(90°)$.

---

[1]In the solar principal plane the polarized reflectance is defined as $R_p = \frac{\pi Q}{E_0 \cos\theta_0}$, where $E_0$ is the extraterrestrial solar irradiance and $\theta_0$ is the solar zenith angle.

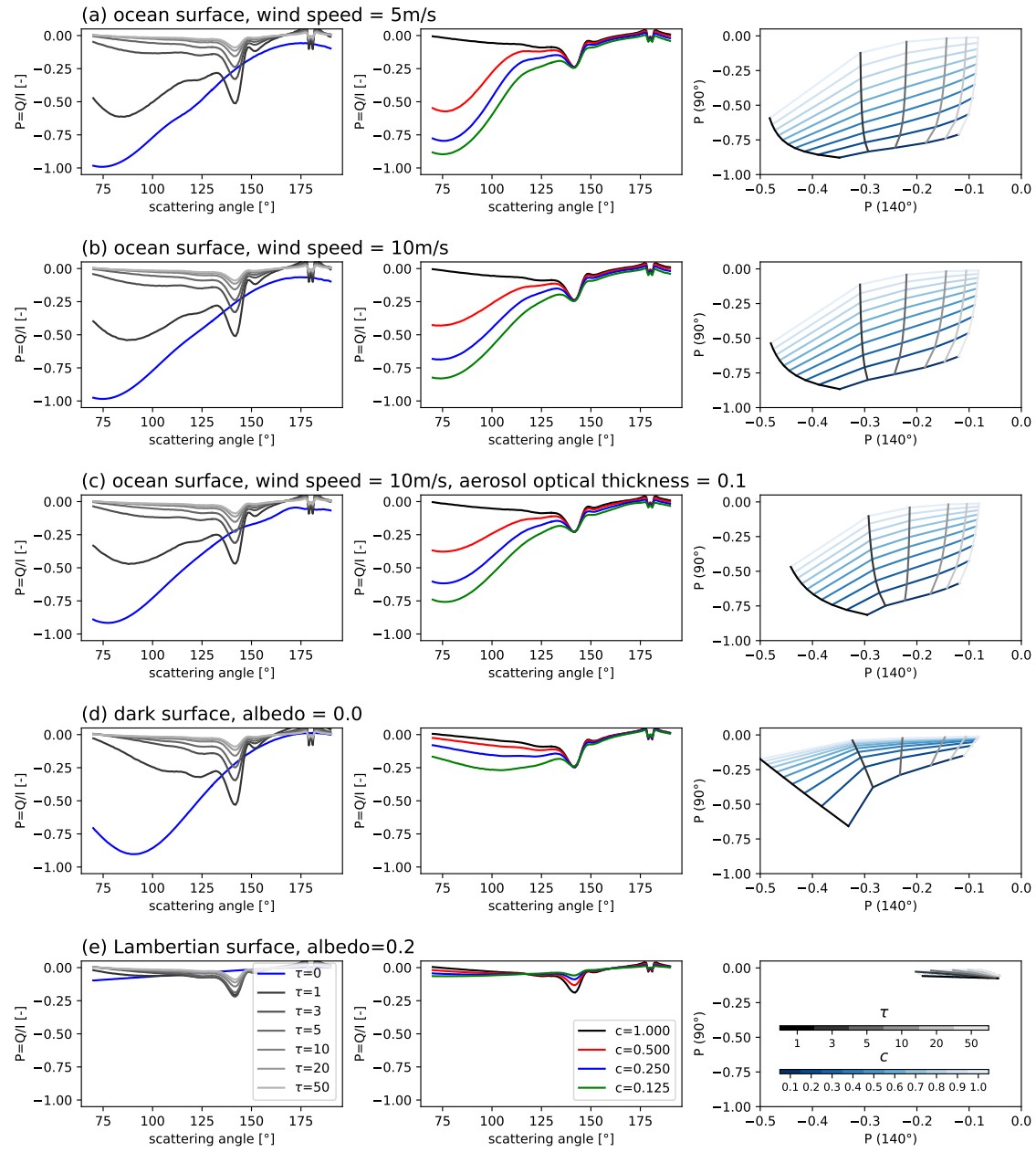

**Figure 3.** Polarized phase curves and retrieval lookup tables for various scenarios. The left figures show simulations for a homogeneous cloud layer and the grey lines correspond to different cloud optical thicknesses. The blue lines correspond to the clear-sky simulation. The middle plots show the polarized phase curves for partially cloudy pixels. The right panels show the corresponding retrieval lookup tables, the line colors correspond to Fig.2.

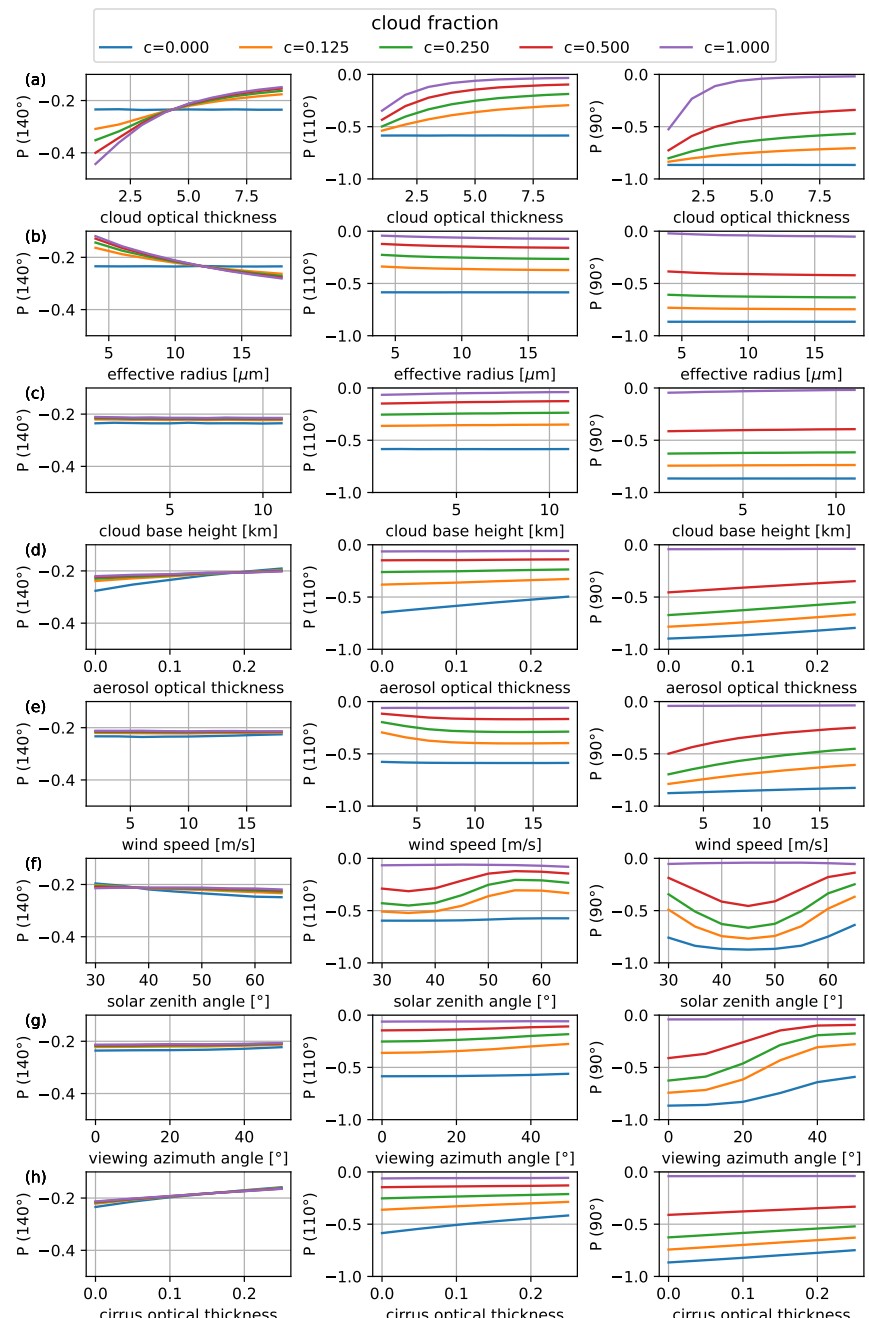

**Figure 4.** Dependencies of degree of polarization $P = Q/I$ at scattering angles of 140° (left), 110° (middle), and 90° (right) on cloud optical thickness, cloud droplet effective radius, cloud base height, aerosol optical thickness, wind speed, solar zenith angle, and viewing azimuth angle. The colors correspond to cloud fractions between 0 and 1 (see legend). All simulations are for an ocean surface.

In scenario (a) the cloud optical thickness $\tau_c$ is varied: as shown before $|P(140°)|$ decreases with increasing $\tau_c$, for all cloud fractions $c$. For $\tau_c \lesssim 4$, the lines corresponding to different cloud fractions separate whereas for larger $\tau_c$, $P(140°)$ is almost independent of cloud fraction. For $P(110°)$ and $P(90°)$ the lines corresponding to different cloud fractions are clearly separated. In scenario (b) the droplet effective radius is varied: Here $|P(140°)|$ increases with increasing effective radius. This implies that in order to retrieve an accurate optical thickness, the retrieval lookup table needs to be generated for the correct effective radius. Therefore, is makes sense to combine the retrieval with an effective radius retrieval based on the cloudbow signature. $P(110°)$ and $P(90°)$ are almost independent of effective radius, therefore the cloud fraction retrieval does not require prior information on droplet size. In scenario (c) the cloud top height is varied, whereas the geometrical thickness of the cloud is kept constant at 1 km. $P(140°)$, $P(110°)$, and $P(90°)$ are almost independent of cloud top height. This is a favorable outcome, suggesting that the retrieval process does not need prior information regarding cloud top height. In scenario (d) the aerosol optical thickness is varied: $P(140°)$ remains constant, while $|P(110°)|$ and $|P(90°)|$ slightly decrease with increasing aerosol optical thickness $\tau_a$. These findings indicate that having prior information on $\tau_a$ would enhance the accuracy of the cloud fraction retrieval. In scenario (e) the wind speed is varied: $P(140°)$ is not impacted by the sun-glint, therefore it is independent of wind speed. $P(90°)$ is independent of wind speed for clear sky pixels and for fully cloudy pixels but not for partially cloudy pixels. Therefore, prior information about wind speed from independent observations should be taken into account. $P(110°)$, at a scattering angle not in the center of the sun-glint, depends much less on the wind speed. This shows, that if there is no prior knowledge of wind speed, it may be better to use $P(110°)$ instead of $P(90°)$ although the polarization signal is weaker. In scenario (f) the solar zenith angle is varied: Again $P(140°)$ is constant but $P(90°)$ and $P(110°)$ vary, because the position of the glint depends on the solar zenith angle. The maximum of the glint is always at the mirror reflection angle, i.e., it moves towards larger scattering angle as the solar zenith angle increases. For a solar zenith angle of 45° the maximum of the sun-glint is at 90° scattering angle. In scenario (g) the viewing azimuth angle is varied, which does not have an impact on $P(140°)$. For $P(90°)$ the impact is quite large, because as the viewing azimuth angle changes, the observing direction moves away from the center of the sun-glint. For $P(110°)$ this dependence is much weaker, as the viewing direction is not close the center of the sun-glint. In order to test, how much the retrieval is influenced by sub-visible cirrus clouds above the liquid clouds we add an optically thin cirrus layer with a cloud top height of 11 km and a geometrical thickness of 1 km (scenario (h)). The effective radius of the crystals is set to 30µm and the parameterisation by (Baum et al., 2014) is applied to obtain the ice cloud optical properties. Generally we see as expected a decrease in degree of polarization with increasing cirrus optical thickness at all scattering angles. This means that with increasing cirrus optical thickness the retrieved liquid water cloud optical thickness will slightly decrease and the retrieved cloud fraction will slightly increase. The sub-visible cirrus layer does not block the glint, therefore the retrieved cloud fraction corresponds approximately to that of the liquid clouds.

In summary, the simulations suggest that the retrieval method is expected to deliver accurate cloud fractions and cloud optical thicknesses over the ocean, provided there is approximate prior information about wind speed and effective radius.

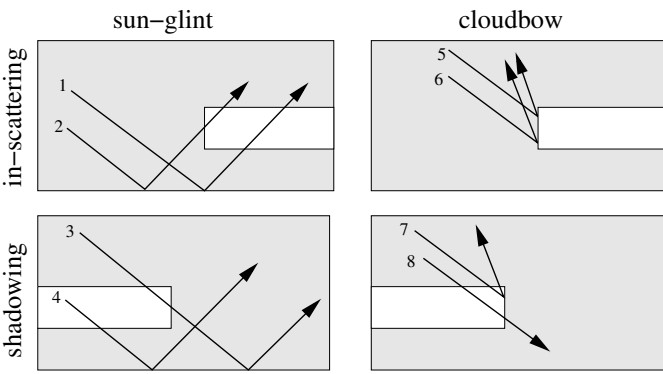

**Figure 5.** Sketch of individual photon paths to explain the basic 3D scattering effects.

## 4 Investigation of 3D scattering effects

In this section we apply the retrieval to the polarized radiances obtained for the 3D model setups defined in Section 2.2.

### 4.1 2D scene with sharp cloud edge

Fig. 5 shows a sketch of the 2D cloud scene including relevant photon paths to explain the 3D effects. Photon 1 reaches the surface without interaction, then becomes highly polarized by reflection at the ocean surface and afterwards passes through the cloud towards the observer. From the observer's perspective there is a cloud in the field of view, but still the sun-glint polarization is partly visible. Photon 2 is similar, but the path through the cloud is shorter at the cloud edge, therefore the polarization is less decreased. Photons 3 and 4 pass the cloud and are reflected by the ocean surface towards the observer. However, much less photons will reach the surface directly compared to clear sky conditions, therefore from the observer's perspective, the degree of polarization in this clear sky pixel will be reduced. Photons 5 and 6 are scattered at the cloud side towards the observer at a scattering angle of 140°, i.e. in the cloudbow region. Due to the geometry of the cloud, additional photons are reflected at the cloud side towards the observer, which results in an increase in intensity and polarization at the cloud edge. For photons 7 and 8 the optical thickness of the cloud is reduced compared to a 1D layer, reducing the probability of scattering in the cloud. Photon 8 passes through the cloud and is eventually scattered in the atmosphere by a molecule.

Fig. 6 shows polarized radiances computed for the 2D cloud as defined in Section 2.2.1 for the in-scattering geometry. The upper panels are for a scattering angle of 140° (cloudbow) and the lower panels for a scattering angle of 90° (sun-glint). The simulations were performed at a spatial resolution of 500 m and averaged to obtain polarized radiances at a spatial resolution of 2.5 km (solid lines shown in Fig. 6). At the cloud edge we average over cloudy and clear-sky sub-pixels, this way we obtain results for cloud fractions of $1/5$, $2/5$, $3/5$ and $4/5$. The dashed lines show the corresponding 1D simulations where we combine clear-sky and cloudy simulation results using the IPA approximation (Eq. 7). In the cloudbow we find as expected an increase of the absolute values of $I$ and $Q$ close to the cloud edge compared to the 1D simulations (see photon paths 5 and 6 in Fig. 5). On the clear-sky side (0–10 km), the magnitude of the degree of polarization $Q/I$ is decreased compared to 1D. On the

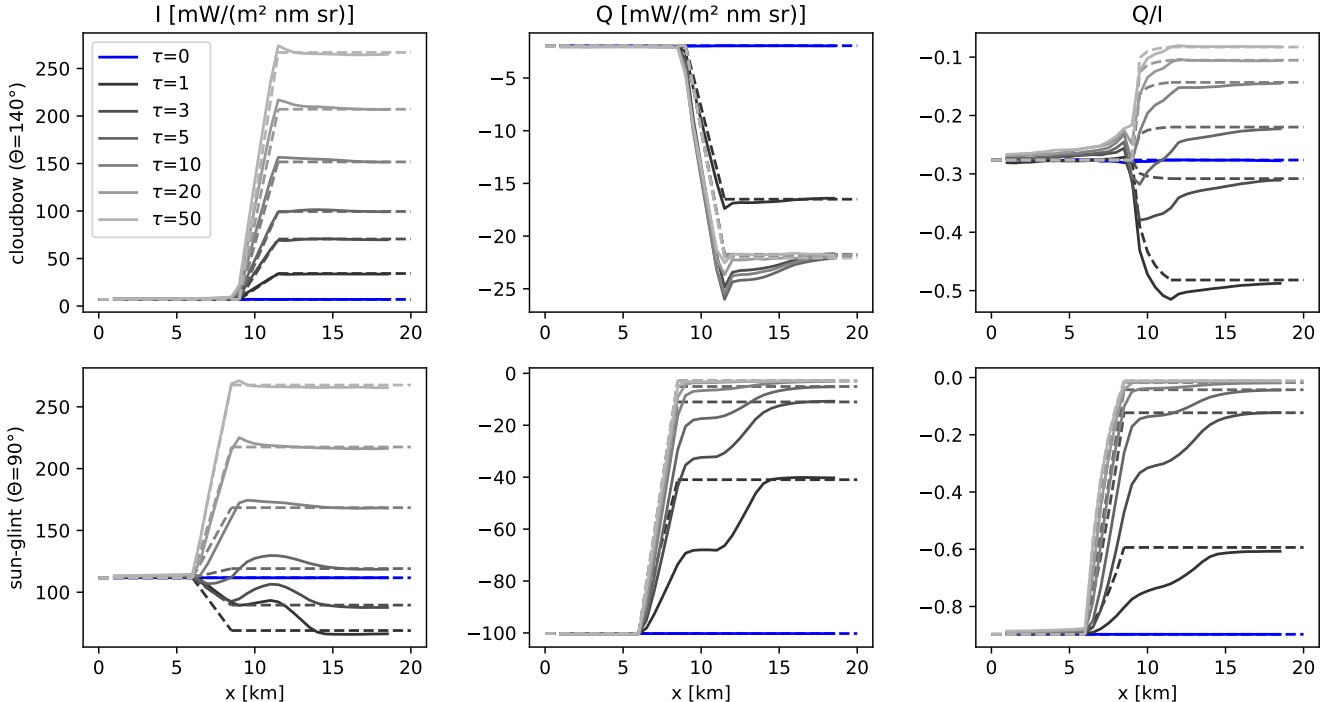

**Figure 6.** In-scattering results for sharp cloud edge: The solid lines show polarized radiances for in-scattering geometry averaged over a region of 2.5 km width. The upper row is for a scattering angle of 140° (cloudbow) and the lower row for a scattering angle of 90° (sun-glint). The dashed lines correspond to 1D simulations combined by the IPA approximation for partly cloud-covered pixels around the cloud edge. The blue lines corresponds to clear sky and the grey lines to cloudy simulations with various optical thicknesses between 1 and 50.

cloudy side (10-20 km) it is increased, which means that the retrieval will underestimate the optical thickness. In the sun-glint, we find the most obvious differences for $Q$ with much higher absolute values compared to the 1D simulations. This is due to photons that reach the surface directly on the clear side of the domain, are then reflected at the ocean surface and traverse the cloud towards the observer at the top of the atmosphere (see photon paths 1 and 2 in Fig. 5). For these cases the retrieval will underestimate the cloud fraction.

Fig. 7 presents the results for the cloud shadow geometry. Here the cloud is located between $x$=0km and $x$=10km. In the cloudbow, the intensity $I$ is decreased at the cloud edge compared to 1D approximations because the optical thickness is apparently smaller (compare photon paths 7 and 8 in Fig. 5). For $Q$ the differences between 1D and 3D are relatively small. The magnitude of the degree of polarization is increased on the cloudy side and decreased on the clear side. In the sun-glint geometry we find very large differences between 3D and 1D simulations in the cloud shadow. Naturally, the intensity $I$ is decreased in the cloud shadow compared to clear sky. The magnitude of the degree of polarization is much smaller in the cloud shadow which can be seen in $Q$ and $Q/I$, obviously because photons cannot reach the surface directly (see photon paths 3 and 4 in Fig. 5). Due to the decreased degree of polarization the retrieval will detect clouds in the cloud shadow region.

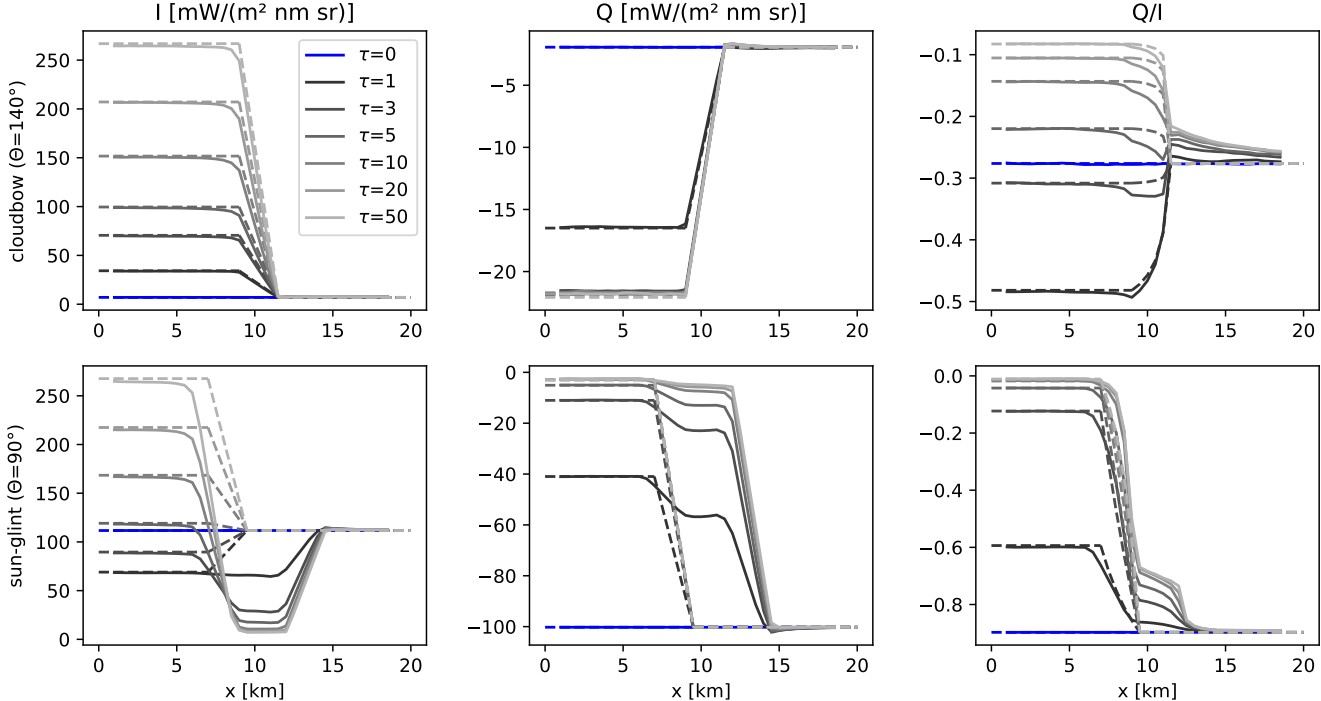

**Figure 7.** Results in the cloud shadow geometry for sharp cloud edge: The solid lines show polarized radiances for shadowing geometry averaged over a region of 2.5 km width. The upper row is for a scattering angle of 140° (cloudbow) and the lower row for a scattering angle of 90° (sun-glint). The dashed lines correspond to 1D simulations combined by the IPA approximation for partly cloud-covered pixels around the cloud edge. The blue lines corresponds to clear sky and the grey lines to cloudy simulations with various optical thicknesses between 1 and 50.

Fig. 8 shows the retrieval lookup table derived in Section 3 including all results of the 3D simulations for the 2D step cloud. For better interpretation we mark all results obtained for in-scattering geometry as circles and all results obtained for shadowing geometry as crosses. In the left panel, the colors of the points correspond to the cloud fraction of the region over which we have averaged (i.e., the input geometrical cloud fraction) and in the right panel the colors correspond to the vertical optical thickness of the cloud. We can check in these plots, for which of the points the retrieval is correct and where the retrieval does not work correctly, just by comparing the colors of the points to the colors of the lines in the lookup table plots. For the in-scattering, we see that fully cloud covered pixels (light blue circles) lie in the lookup-table between cloud fractions of 0.8 and 1.0. Dark blue crosses correspond to the points in the clear region, but many of those points lie on a vertical line along the $\tau$=3 isoline, these points correspond to simulations in the cloud shadow where a cloud is erroneously detected. The points corresponding to optical thickness of 1 (black points in right figure) lie outside the lookup table because in the cloudbow the magnitude of the degree of polarization in 3D is larger than in 1D, for in-scattering and shadowing.

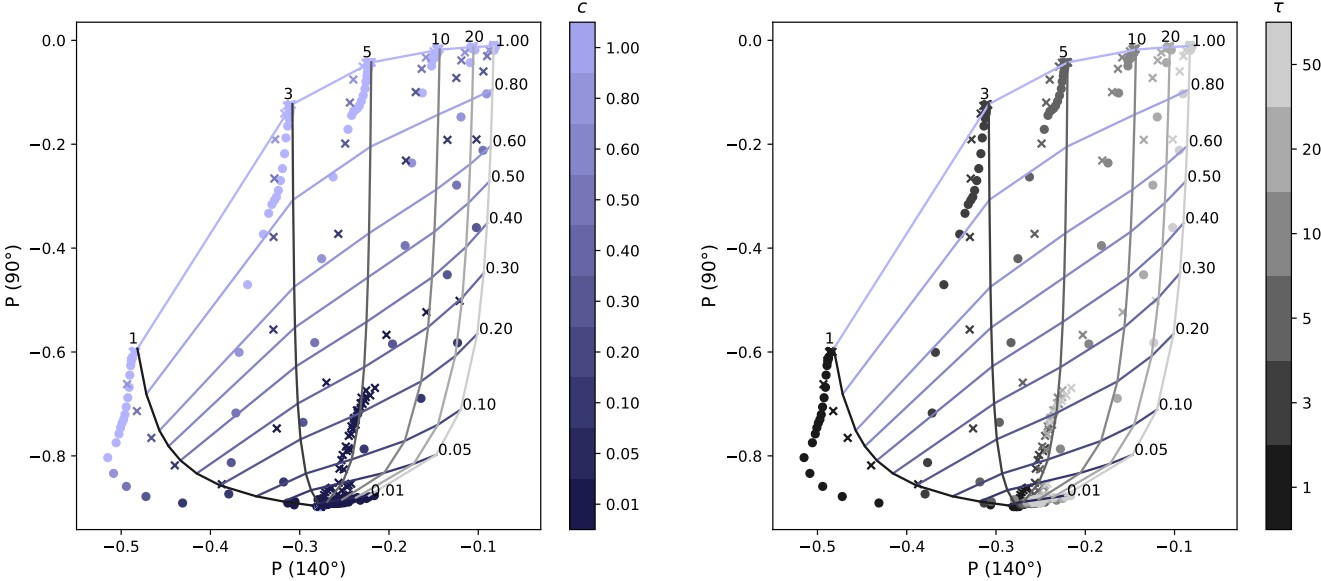

**Figure 8.** Retrieval lookup table derived in Section 3 (Figure 2) including the results of the 3D simulations for the 2D cloud with sharp edge. Circles are for in-scattering and crosses for shadowing. In the left panel the colors of the points correspond to the input cloud fraction and in the right panels the colors correspond to the input vertical optical thickness.

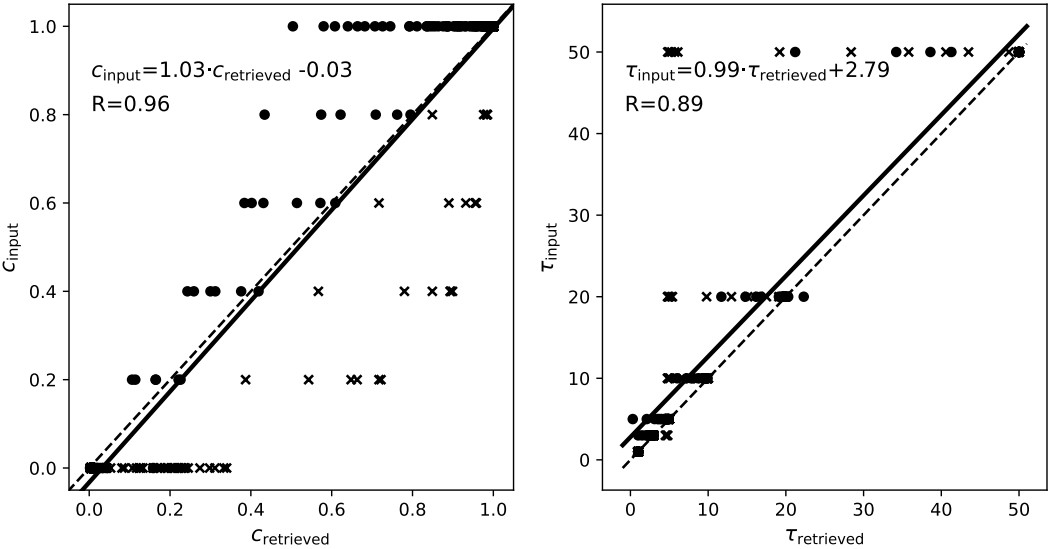

**Figure 9.** Scatter plots of the retrieved versus the input cloud fractions (left) and optical thicknesses (right). The circles correspond to in-scattering and the crosses to shadowing. In the right panel points with a retrieved cloud fraction smaller than 0.1 have been filtered out. The solid black line shows the linear regression line and the dashed line is the one-one-line. The equation of the linear regression lines, and the correlation coefficients $R$ are included in the figure.

Fig. 8 shows that 3D scattering effects cause significant biases in the retrieval results for cloud fraction and optical thickness. In order to validate these biases more quantitatively we show the retrieved data against the input data as scatter plots in Fig. 9. In the scatter plots, the data points are located on horizontal lines because the input cloud fraction and optical thickness values are discretized. The left panel shows the retrieved cloud fraction versus the input cloud fraction (vertical geometrical cloud fraction). Dots marked by filled circles correspond to the values obtained for the in-scattering geometry. As already discussed above, we find an underestimation of the cloud fraction, because in this geometry the degree of polarization for the 2D cloud scene is higher than for the corresponding 1D cloud layer. For the shadowing geometry (marked by crosses) the cloud fraction is overestimated because the degree of polarization is decreased in the shadow. Under- and overestimation can both become quite large for the extreme case of a sharp cloud edge. For example, for an input cloud fraction of 0.4, the retrieved cloud fractions are in the range between 0.2 and 0.9. The slope of the linear regression line is 1.03 and the correlation coefficient is 0.96. These values show that although there is a large spread in the retrieved cloud fraction, the biases by in-scattering and shadowing almost cancel each other in this particular case. The right panel shows the scatter plot for the optical thickness retrieval. Here we filtered out the points for which the retrieved cloud fraction is smaller than 0.1, because for very small cloud fractions the retrieval becomes insensitive to optical thickness. Generally, we find a good correlation between input and retrieved optical thickness, but there is a significant bias towards too small retrieved optical thicknesses (the regression line is shifted to the left compared to the one-one line). This is the expected result, because we have seen for in-scattering and shadowing, that the absolute value of the degree of polarization is larger in the 3D results than in the corresponding 1D results. The underestimation of the cloud optical thickness due to the neglect of 3D cloud scattering has also been observed for other cloud optical thickness retrieval methods (e.g., Zinner et al., 2010; Alexandrov et al., 2024). Note that we still look at the extreme case of a very sharp cloud edge, for which we expect very strong 3D scattering effects.

## 4.2 Randomly distributed box clouds

In the following, we present the results for the randomly distributed box clouds which shall resemble a shallow cumulus field (see Fig. 10). In the cloudbow, the intensity $I$ is only weakly influenced by 3D scattering effects, i.e. the solid lines for the broken cloud fields lie on top of the dashed lines showing the IPA calculations. $|Q|$, which saturates quickly for $\tau \geq 3$, is slightly larger for 3D compared to IPA. This difference is enhanced in the degree of polarization shown in the right panels. In the sun-glint, the intensity $I$ is smaller for 3D than for IPA due to cloud shadows. Note that the apparent noise of the curves is due to the random cloud field generation, not due to Monte Carlo noise, which is $< 1\%$. $|Q|$ is decreased significantly in 3D compared to 1D, also due to cloud shadowing. Since the effect of shadowing is the same for $I$ and $Q$ it partly cancels out in the degree of polarization, but still, $|P|$ is smaller in the 3D compared to 1D. This effect will cause a systematic overestimation of the retrieved cloud fractions.

Fig. 11 shows the retrieval lookup table including the results of the 3D simulations for the randomly distributed box clouds. The points corresponding to constant optical thickness values align next to the isolines of constant optical thickness, but shifted to the left. This shows again, that the neglect of 3D scattering yields an underestimation of cloud optical thickness. The colors of

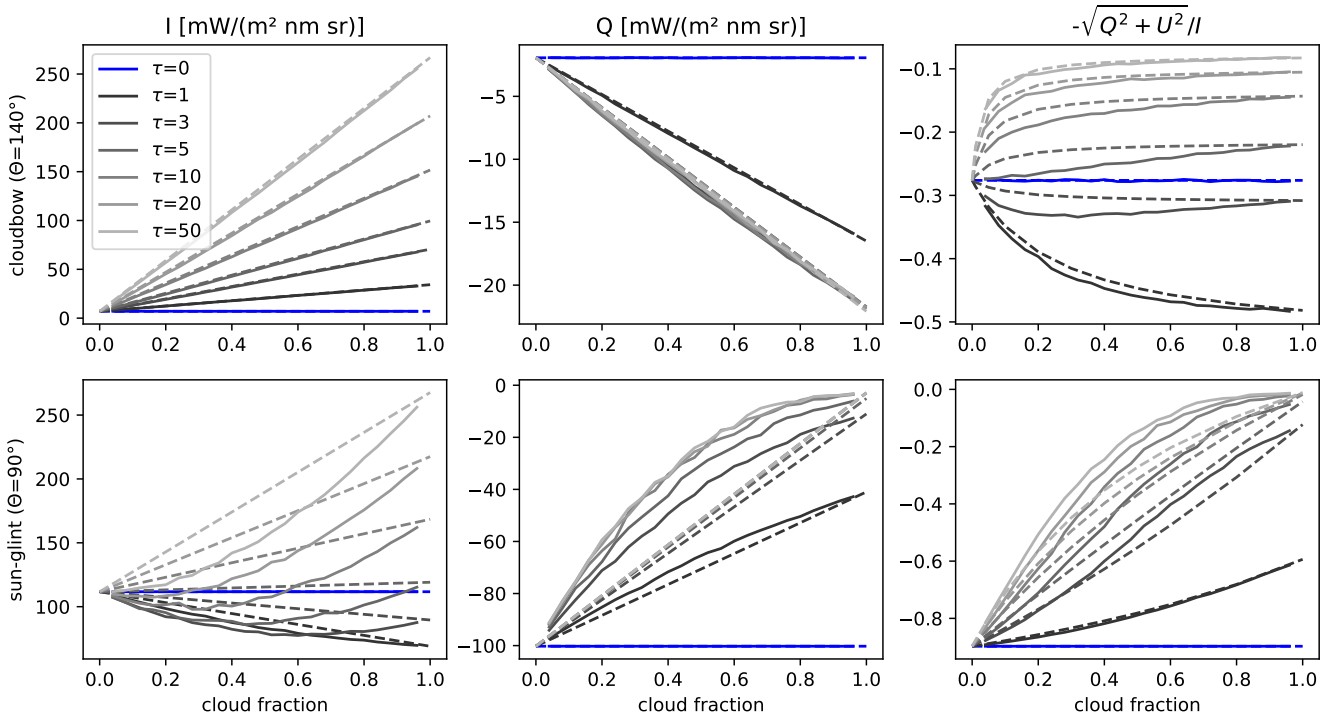

**Figure 10.** Polarized radiance ($I$ and $Q$) and degree of polarization $P$ as a function of cloud fraction. The solid lines are for randomly distributed box clouds and the dashed lines are IPA calculations. The spatial resolution of the individual grid boxes is $500 \, \mathrm{m}^2$ and the vertical geometrical thickness of the cloud boxes is $100 \, \mathrm{m}$.

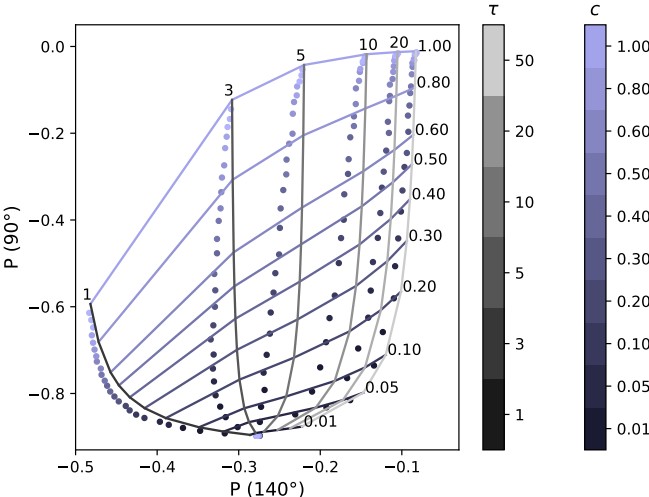

**Figure 11.** Lookup table including the results of the 3D simulations for the randomly distributed box clouds. The colors of the dots correspond to the input geometrical cloud fraction.

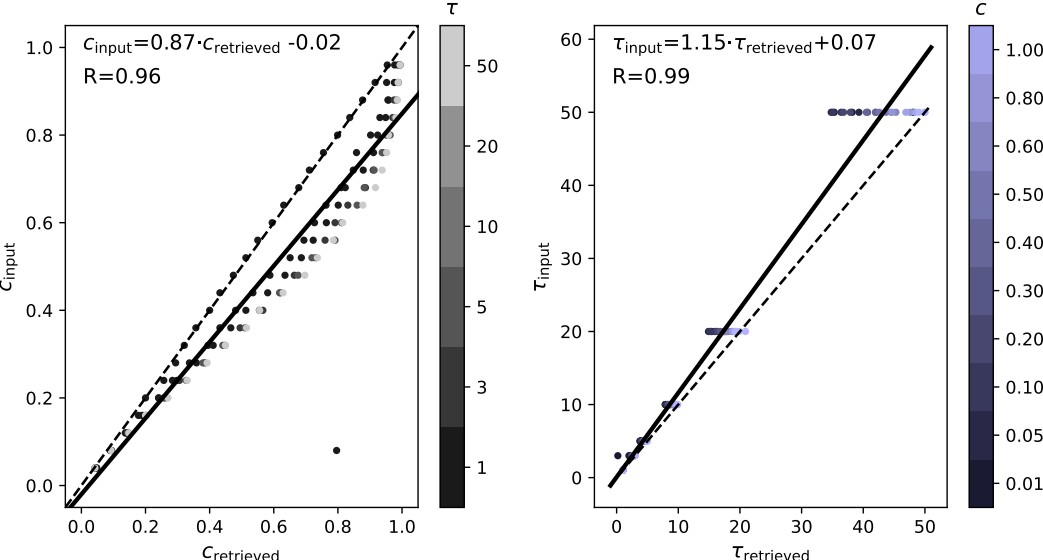

**Figure 12.** Retrieval results for randomly distributed shallow clouds. The left panel shows the cloud fraction retrieval. The color of the dots correspond to the input optical thickness. The right panel shows the optical thickness retrieval and the color of the dots correspond to the input cloud fraction. The solid black line shows the linear regression line and the dashed black lines are the 1:1 lines. The equation of the linear regression line, and the correlation coefficient $R$ are included in the figures.

the dots correspond to the geometrical cloud fraction of the model input. They roughly match the colors of the lines, indicating that the cloud fraction retrieval will yield reasonable results.

More quantitatively, Fig. 12 presents the retrieval results as scatter plots. The left panel shows the cloud fraction retrieval. As 355 expected, we find a systematic overestimation of the retrieved cloud fraction, because the cloud shadows decrease the degree of polarization in the sun-glint region. The error becomes larger for larger input optical thicknesses as the colors of the dots indicate. Only the points corresponding to a cloud fraction of 1 (fully cloudy) lie exactly on the one-one line. Note that for our specific case of shallow broken cloud fields, we do not get an underestimation of cloud fraction due to in-scattering. The right panel shows the vertical optical thickness retrieval which is biased towards too small values, consistently with the cloud 360 edge case. The underestimation increases with decreasing cloud fraction as indicated by the color of the dots in the right panel. There is one outlier for which the retrieved optical thickness is very small and the cloud fraction is then largely overestimated. Overall, we obtain a very good correlation between the retrieval results and the input data for both, the cloud fraction ($R$=0.95) and the cloud optical thickness ($R$=0.99).

All results presented in this section are for a solar zenith angle of 50°. We have run the same simulations for solar zenith 365 angles of 30° and 70° and found a similar performance of the retrieval. Generally, with increasing solar zenith angle the overestimation of cloud fraction increases. The underestimation of cloud optical thickness on the other hand decreases with increasing solar zenith angle.

## 5   Testing the retrieval method on high spatial resolution aircraft data

In this section we test the retrieval method using airborne observations taken by the spectrometer of the Munich Aerosol Cloud Scanner (specMACS) on the HALO (High Altitude and LOng Range research) aircraft during the EUREC[4]A (ElUcidating the RolE of Cloud-Circulation Coupling in ClimAte) measurement campaign (Stevens et al., 2021).

The specMACS instrument (Ewald et al., 2016) consists of two hyperspectral line cameras covering the spectral range from 400 to 2500 nm and two identical polarization-sensitive imaging cameras (Pörtge et al., 2023; Weber et al., 2023). The two polarization-sensitive cameras have a combined maximum field of view of about $\pm 91° \times \pm 117°$ (along track $\times$ across track). This results in a horizontal resolution of 10–20 m at ground when the flight altitude is around 10 km. The cameras take images at an acquisition frequency of 8 Hz. The sensors include on-chip directional polarizing filters which allow to measure the intensity and the linear polarization, i.e. the Stokes vector components $I$, $Q$, and $U$. These are geometrically and radiometrically calibrated (Weber et al., 2023). The central wavelengths (bandwidths) of the three channels of specMACS are approximately 621 nm (66 nm), 547 nm (117 nm) and 468 nm (82 nm). In the following we use data that were measured during the EUREC[4]A field campaign which took place in January and February 2020 with base in Barbados. All observations during the EUREC[4]A campaign were taken over ocean.

We selected six scenes including shallow cumulus clouds observed on $28^{th}$ of January 2020. These contain various cloud fractions, from almost clear to almost fully cloud covered. The size of the scenes is approximately $2.5 \times 2.5$ km$^2$ equivalent to the pixel size of, e.g., HARP2 and the spatial resolution of the data is approximately 10 m. The images corresponding to the selected scenes $s1$ to $s6$ are presented in Fig. 13, where each cloud scene is shown at scattering angles around 110° (left) and at scattering angles around 140° (right). Note that in the left images the ocean surface is clearly brighter than in the right images due to the sun-glint whereas the brightness of the clouds is similar in both images. Across an image the scattering angle is not constant. Since we require observations at particular scattering angles we use the geo-localization as described in Kölling et al. (2019) and Pörtge et al. (2023) in order to obtain all pixels in the chosen region for a given scattering angle. Pixels obtained using this method are averaged to get the Stokes vector of a selected scene for a given scattering angle. Combining all scattering angles results in the phase curves which are shown in Fig. 14 for the selected scenes for the red channel of specMACS centered at 621 nm. Since the observations are not taken exactly in the solar principal plane, the $U$-component of the Stokes vector becomes non-zero and we use Eq. 6 to calculate the signed degree of linear polarization. As expected we find that $|P|$ decreases with increasing cloud amount in the scenes ($s1$ is the scene including very few clouds and $s6$ is almost fully cloud covered).

In order to retrieve cloud fraction and cloud optical thickness from specMACS data we generate a lookup table for the specific time and conditions when the observations were taken. We performed monochromatic simulations for the center wavelength of the red channel (621 nm). The solar zenith angle was 46.5° and we know that the total column aerosol optical thickness on the particular day was about 0.08 (Chazette et al., 2022). For simplicity we use a typical wind speed of 5 m/s, which is in agreement with Special Sensor Microwave Imager Sounder (SSMIS) observations (Wentz et al., 2012). We also applied standard values for the water cloud droplet size distribution: an effective radius of 10 μm and an effective variance of

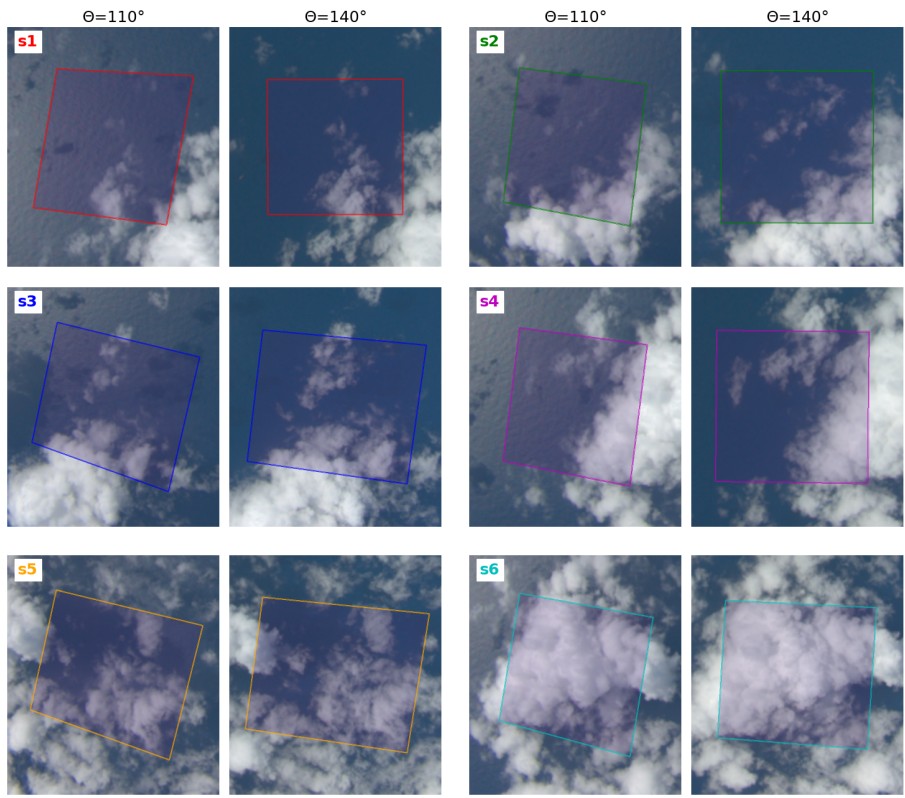

**Figure 13.** Six selected cloud scenes $s1$ to $s6$ observed by specMACS over ocean. The size of the shaded area in each image is approximately $2.5 \times 2.5 \text{ km}^2$. For each of the scenes the cloud field is observed from different viewing directions during the overflight. In the left images of each scene, the average scattering angle over the shaded region is $110°$ and in the right images it is $140°$.

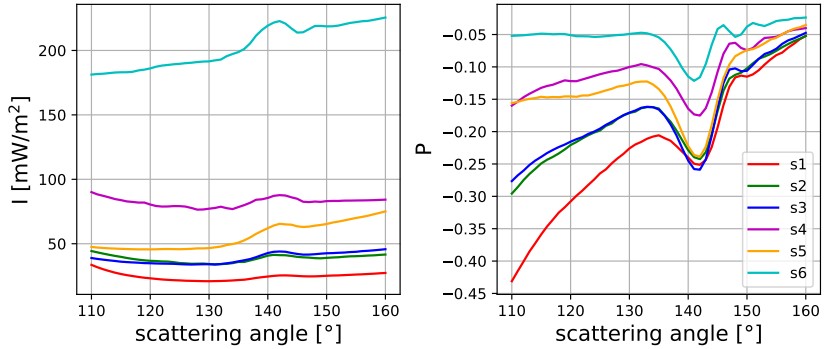

**Figure 14.** Phase curves of $I$ and $P$ observed by specMACS in the red channel centered at 621 nm for the selected scenes which are shown in the images of Fig. 13.

0.1. Results of the cloudbow retrieval (Pörtge et al., 2023) show that these values are realistic (compare Table 1). As we have demonstrated in the sensitivity analysis in Section 3.2, inaccurate assumptions on cloud-size distribution parameters produce errors in the retrieval of cloud optical thickness while the cloud cover retrieval is not much affected. For the selected scenes,

we generated lookup-tables using the cloud size distribution parameters from the cloudbow retrieval in addition. Of course, it would make sense to combine the method with further retrieval algorithms, e.g., with the simultaneous aerosol and ocean glint retrieval by Knobelspiesse et al. (2011). When accurate a priori information about wind speed and aeorosol optical thickness is included, one should also take into account the filter function of the instrument rather than running monochromatic simulations to generate the lookup-table. These improvements are not necessarily needed to demonstrate the method for a few specific

cases, which is the purpose of this study. With specMACS we also can not obtain scattering angles of 140° and 90° for the same region, therefore we use 140° and 110° to generate the lookup table shown in Fig. 15. The lookup table based on 140° and 110° that we use for the specMACS data is slightly more tilted compared with the lookup table based on 140° and 90° that was mainly presented until here (Fig. 2). Nevertheless, the lookup table for the specMACS data still appears to be well suited to apply the method. The colored dots plotted on top of the lookup table correspond to the observational data of the six

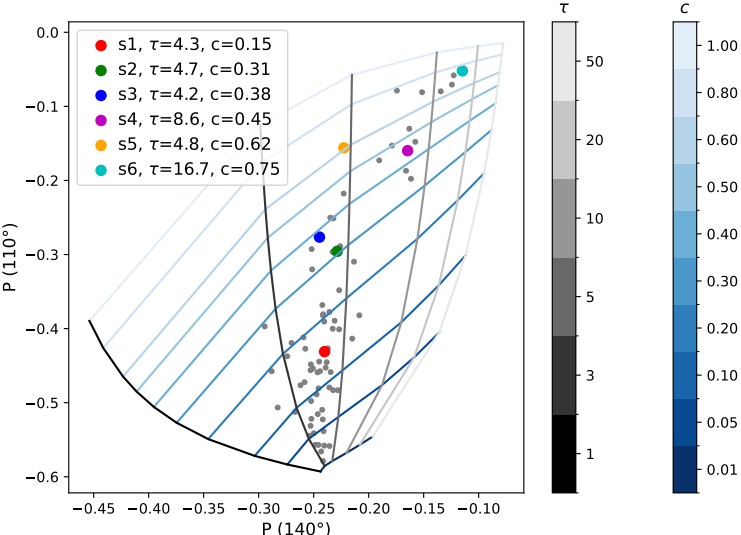

**Figure 15.** Retrieval lookup table and observational data. The colored dots represent the observed values of the six selected scenes shown in Fig. 13. The legend includes the retrieved optical thickness and cloud fraction values for those points. The grey dots are all data points gathered during one research flight on $28^{th}$ January 2020 (observations averaged to 2.5 km spatial resolution).

selected scenes. The grey dots show all other data points that were measured with specMACS. Using a bi-linear interpolation between the simulated grid points of the lookup table we retrieve the values for cloud optical thickness $\tau$ and cloud fraction $c$ as included in the legend of Fig. 15.

We also employed the cloud detection method outlined in Pörtge et al. (2023) on the images of the selected scenes to determine the geometrical cloud fraction of the scenes at the original (high) resolution of the images. The method is based on

the algorithm described in Otsu (1979) which determines a threshold to separate the pixels of an image into two classes (here: cloudy versus non-cloudy) based on a brightness histogram. Such threshold based cloud detection algorithms often struggle with the bright sun-glint reflection. Our algorithm uses the parallel component of polarized light in which the reflectance of the sun-glint is reduced which in turn reduces the number of incorrect classifications in (clear-sky) sun-glint areas. The algorithm distinguishes between cloud-free, low to medium cloud coverage, and high cloud coverage. For cloud-free scenes it uses the data of the blue channel, for scenes with low/medium cloud coverage the data of the red channel, and otherwise the normalized red ($r$) to blue ($b$) ratio ($\mathrm{nrbr} = (b - r)/(b + r)$). This procedure was found by tuning the cloud mask for many different scenes (both over land and over water). Figure 16 shows two example measurements with contourlines of the calculated cloud mask in light-green. This figure illustrates that the algorithm correctly identifies the large cloud structures but misses some of the smaller and optically thinner clouds. Until now, the cloud detection method was mainly used to identify measurements that are suitable for the microphysical cloudbow retrieval of Pörtge et al. (2023). Therefore, the algorithm was tuned to minimize the amount of falsely classified clouds (e.g. due to the sun-glint).

The colored boxes in Fig. 16 indicate the positions of the six scenes ($s1$ to $s6$) from which the (pixel-based) cloud fraction is calculated. It should be noted that using this approach, the cloud fraction depends on the viewing angle but this dependence is relatively small for shallow clouds. Again, please note the difference between the two approaches: The threshold-based results correspond to a pixel-by-pixel cloud fraction and the lookup table allows a retrieval of the cloud fraction, which is independent of spatial resolution and does not rely on thresholds.

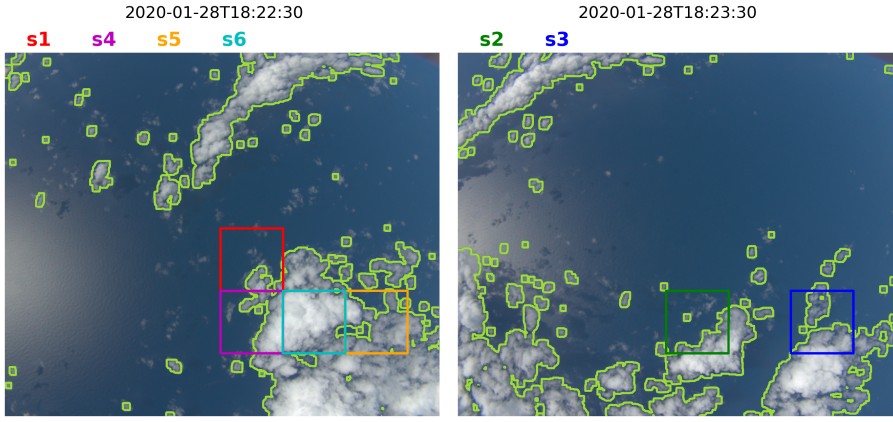

**Figure 16.** RGB images at two times (left: 18:22:30 and right: 18:23:30 UTC) with cloudmask as light-green lines. The regions of the selected scenes $s1$ to $s6$ are indicated as colored boxes.

Results for the selected scenes are included in Table 1: $\tau_1$ and $c_1$ are, respectively, the optical thickness and the cloud fraction obtained from the lookup-table retrieval using constant cloud size distribution parameters as input. $\tau_2$ and $c_2$ are obtained when the retrieval is combined with the cloudbow retrieval, taking into account the retrieved effective radius $r_{\mathrm{eff}}$ and effective variance $v_{\mathrm{eff}}$, which are also given in the table. $c_3$ is the (threshold-based) cloud fraction, which is defined as

| retrieval method | $P(110°)/P(140°)$ | | $P(110°)/P(140°)$ | | cloud detection | cloudbow | |
|---|---|---|---|---|---|---|---|
| scene | $\tau_1$ | $c_1$ | $\tau_2$ | $c_2$ | $c_3$ | $r_\text{eff}$ | $v_\text{eff}$ |
| s1 | 4.3 | 0.15 | 3.5 | 0.17 | 0.15 | 8.6 | 0.23 |
| s2 | 4.7 | 0.31 | 4.4 | 0.33 | 0.28 | 9.7 | 0.22 |
| s3 | 4.2 | 0.38 | 4.5 | 0.37 | 0.52 | 10.7 | 0.10 |
| s4 | 8.6 | 0.45 | 9.1 | 0.43 | 0.43 | 10.7 | 0.09 |
| s5 | 4.8 | 0.62 | 4.7 | 0.62 | 0.77 | 9.8 | 0.15 |
| s6 | 16.7 | 0.75 | 31.1 | 0.70 | 0.98 | 14.5 | 0.06 |

**Table 1.** Retrieval results for the selected scenes. $\tau_1$ and $c_1$ are the cloud optical thickness and the cloud fraction derived from the degree of linear polarization at 110° and 140° assuming constant values of $r_\text{eff}$=10 µm and $v_\text{eff}$=0.01. $\tau_2$ and $c_2$ correspond to retrieval results taking into account $r_\text{eff}$ and $v_\text{eff}$ from the cloudbow retrieval (last two columns). $c_3$ is the fraction of cloudy pixels in the scene which is determined using a cloud detection algorithm on the high spatial resolution data.

the ratio of the number of cloudy pixels and the total number of pixels in the high spatial resolution image. For scene $s1$ we obtain $\tau_1$=4.3 and $c_1$=0.15 from the lookup-table using constant cloud size distribution parameters. When the retrieved size distribution parameters from the cloudbow retrieval are used, the optical thickness is reduced to $\tau_1$=3.5 and the cloud fraction is slightly increased to $c_2$=0.17. The threshold-based cloud fraction for this case is $c_3$=0.15, which agrees very well to the lookup-table based results. For scene $s2$ the lookup table yields optical thicknesses of $\tau_1$=4.7 and $\tau_2$=4.4 and cloud fractions of $c_1$=0.31 and $c_2$=0.33, which are slightly larger than the threshold based cloud fraction $c_3$=0.28. For scene $s3$ the threshold based cloud fraction is with $c_3$=0.52 much larger than for $s2$ which is not so clearly visible when comparing the images in Fig. 13. Looking at the contours of the cloud mask in Fig. 16, we see that for $s3$ a large region including thin clouds is marked as cloudy, whereas in $s2$, several thin clouds are not detected. The lookup-table based cloud fraction values are smaller ($c_1$=0.38 and $c2$=0.37) and therefore only slightly larger than those derived for $s2$. The cloud optical thickness values ($\tau_1$=4.2 and $\tau_2$=4.5) are also similar to $s2$. For $s4$, all retrieved cloud fractions agree well ($c_1$=0.45, $c_2$=0.43, and $c_3$=0.43). The optical thicknesses are larger than in the previous scenes ($\tau_1$=8.6 and $\tau_2$=9.1). For scene $s5$ the lookup-table retrievals yield the same cloud fraction of $c_{1,2}$=0.62. The threshold based cloud fraction is significantly higher with $c_3 = 0.77$. The cloud optical thickness for $s_5$ is relatively small ($\tau_1$=4.8 and $\tau_2$=4.7). For all scenes discussed so far, the error due to the assumption of constant size distribution parameters was relatively small. This is different in the last scene $s6$, for which the cloudbow retrieval yields a significantly larger effective radius of 14.5 µm. Here, the retrieved optical thickness is almost doubled ($\tau_1$=16.7 and $\tau_2$=31.1) when the correct size distribution is used to generate the lookup table. The impact on the cloud fraction retrieval is relatively small ($c_1$=0.75 and $c_2$=0.70). The cloud detection algorithm classifies almost all pixels in $s6$ as cloudy, resulting in a cloud cover of $c_3$=0.98. Looking at the images in Fig. 13 it seems that there are some clear-sky areas in the lower right part of the images which are classified as cloudy by the cloud detection algorithm. However, visually the cloud fraction looks larger than 0.75, so we can not clearly conclude, which of the methods performs better for the specific scene $s6$.

The grey dots included in the lookup table in Fig. 15 correspond to all specMACS observations taken during a one hour period of the research flight on $28^{th}$ of January 2020, for which the viewing azimuth angle is not more than $40°$ away from the principal plane, making sure that the sun-glint is contained. In this range, $P(110°)$ does not depend much on viewing azimuth angle (compare Fig. 4). All dots are within the lookup table grid which means that the retrieval yields values for each observation. The retrieved cloud fractions and optical thicknesses are shown as histograms in Fig. 17. The EUREC⁴A campaign

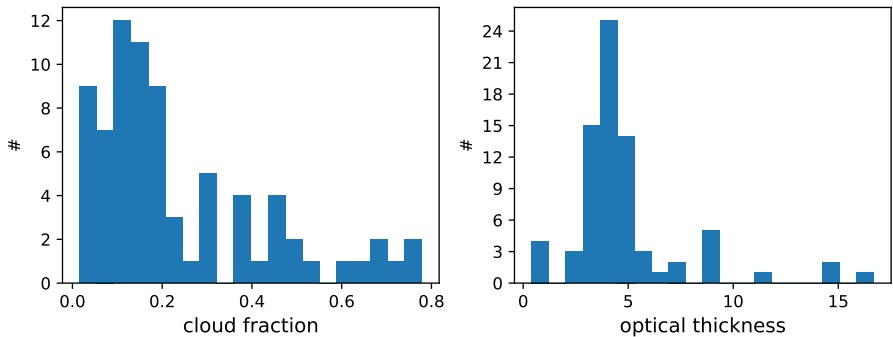

**Figure 17.** Histogram of retrieved cloud fractions and optical thicknesses using the lookup table method.

focused on shallow cumulus clouds which were also observed on $28^{th}$ of January 2020. The optical thickness of this type of clouds is usually quite small which can also be seen in the cloud optical thickness histogram that we obtain for the particular flight, showing an optical thickness below 10 for the majority of the points. A large range of cloud fraction values between 0 and 0.8 is covered by the observations when the cloud detection algorithm is evaluated over areas of $2.5×2.5 \, \text{km}^2$. The majority of points have relatively low cloud fractions (i.e. smaller than 0.2). In Fig. 18, the retrieved cloud fractions obtained through the lookup-table method are plotted on the $x$-axis versus the retrieved cloud fractions obtained through the threshold method on the $y$-axis. The linear regression line has a slope of 0.87, and the correlation coefficient is 0.79. The scatter plot demonstrates that the threshold-based retrieval often results in larger cloud fractions compared to the lookup table-based retrieval, a pattern consistently observed in the selected scenes $s5$ and $s6$. The strong correlation confirms that the straightforward lookup table retrieval method provides reasonable cloud fraction values for the entire flight's observations.

## 6 Conclusions and outlook

We have presented an innovative approach to retrieve cloud fraction and optical thickness of liquid clouds over ocean from multi-angle polarization observations over ocean. This retrieval method could be a valuable addition, for instance, to the cloud retrieval chain for the PACE mission including the polarimeters HARP2 and SPEXone. Here it should be combined with other retrieval methods to obtain required a priori information on aerosol optical thickness, wind speed, and cloud droplet size distribution. Given the typical spatial resolution of upcoming polarized satellite-based measurements of approximately 2-3 km, this technique becomes particularly valuable for acquiring sub-pixel information about clouds. Unlike most other

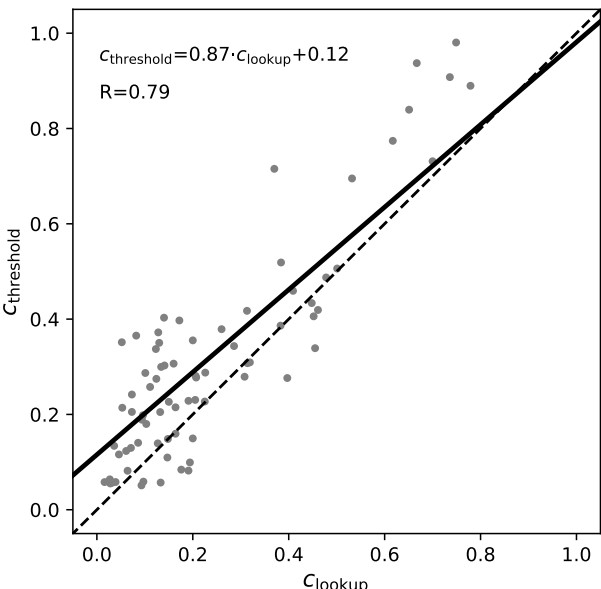

**Figure 18.** Scatter plot of the retrieved cloud fractions: $c_{\mathrm{lookup}}$ is the cloud fraction derived by the lookup table method from the polarized reflectance at two viewing angles. $c_{\mathrm{threshold}}$ is the ratio of cloudy pixels to the total number of pixels within an area of size $2.5 \times 2.5 \ \mathrm{km}^2$ contained in a specMACS image with a spatial resolution of about $10 \ \mathrm{m}$. The equation of the linear regression line (black line) is included in the figure together with the regression coefficient $R$.

cloud fraction retrieval methods, our approach does not depend on the spatial resolution of the observations and it also does not
require any thresholds for cloud detection. The fundamental principle of this method lies in the angular polarization patterns generated by the interaction of radiation with the liquid water clouds and water surface. Specifically, cloud scattering generates polarization at cloudbow scattering angles, while ocean surface reflection results in the strongly polarized sun-glint around the mirror reflection angle. Consequently, by analyzing the angular polarization pattern, we can separate the contributions of clouds and the surface to the observed polarized intensity. Our sensitivity study reveals that the complete angular pattern of polarized
intensity is not required. Instead, two specific scattering angles – one within the cloudbow and one within the sun-glint – are sufficient. Utilizing a 1D vector radiative transfer code, we generated the retrieval lookup tables that include the simulated degree of polarization at $140°$ and $90°$ scattering angles for various cloud optical thicknesses between 0 and 50. To calculate polarized intensities for partially cloudy pixels we employ the independent pixel approximation. We also generated lookup tables using scattering angles of $140°$ and $110°$ and again obtained a well-separated lookup-table grid. This demonstrates that
it is not important to choose exact scattering angles to set up the retrieval, one only has to make sure, that one angle includes the cloudbow and the second angle a part of the sun-glint.

    We examined the influence of cloud droplet size, cloud top height, aerosol optical thickness, and wind speed on the retrieval accuracy. We found that the retrieval results are almost independent of cloud top height. The cloud fraction retrieval slightly

depends on aerosol optical thickness and on wind speed, therefore prior knowledge on these parameters is advantageous. The retrieval of cloud optical thickness depends on droplet radius. Hence, deriving this parameter, for instance, through methods like the cloudbow retrieval, can significantly enhance accuracy.

We investigated the impact of 3D cloud scattering by generating synthetic observations for simple cloud cases. The first is a 2D scene, which is clear in one half of the domain and cloudy in the other half. At such a sharp cloud edge we expect strong 3D effects, shadowing on one side and in-scattering on the other side. We find that shadowing leads to a systematic overestimation of the cloud fraction, whereas in-scattering leads to a systematic underestimation. The cloud optical thickness is generally underestimated due to the neglect of 3D cloud scattering. In a second setup, we generated random cloud fields consisting of box clouds with various cloud fractions. This setup resembles a shallow cumulus cloud field and it allows us to systematically investigate the impact of 3D cloud scattering as a function of cloud cover. For this setup we found that the retrieval systematically overestimates the cloud fraction and underestimates the cloud optical thickness, similar to the cloud edge case. The cloud fraction underestimation is not observed in this scenario. The value of the bias depends on many characteristics of the cloud field, in particular the horizontal and vertical distribution of the clouds. In addition, the geometrical thickness of the clouds plays a significant role since it directly determines the size of the cloud shadows.

We tested the method using specMACS observations conducted aboard the HALO aircraft during the EUREC$^4$A campaign, focusing on shallow cumulus clouds over the ocean. Operating at a high spatial resolution of approximately 10-20 m, spec-MACS data was averaged over domains of approximately $2.5 \times 2.5$ km$^2$ to emulate satellite observations. Subsequently, the retrieval method was applied to the resulting degree of polarization at scattering angles of 140° and 110°. The retrieved optical thicknesses were typically below 10, a realistic range for the observed cloud type, and cloud fractions in the range between 0 and 0.8 were retrieved. We selected six scenes, representing low, medium, and high cloud fractions. We analyzed the high-resolution images to visually verify the consistency between the retrieved cloud fraction and the images. Additionally, we applied another cloud fraction retrieval method based on (automatically determined) intensity thresholds (Pörtge et al., 2023; Otsu, 1979). We find a good correlation between the results of the two cloud fraction retrieval methods, providing further evidence of the performance of the straightforward lookup-table retrieval approach.

In a subsequent study, we intend to further evaluate the accuracy of the retrieval method using synthetic data based on more realistic clouds generated using a Large Eddy Simulation (LES). As we know the true optical thicknesses and cloud fractions for the synthetic dataset, assessing the retrieval accuracy becomes straightforward by comparing the obtained results with the model input. This approach will enable us to quantify the effects of 3D cloud scattering, including in-scattering and shadowing, on the retrieval results. We have successfully applied this methodology to validate the accuracy of the cloud microphysics retrieval (Volkmer et al., 2023) and to investigate the impact of 3D cloud scattering on trace gas retrievals (Emde et al., 2022; Yu et al., 2022; Kylling et al., 2022). In this paper we have presented 3D effects based on simplified cloud scenes. For the case with the cloud edge, we found large underestimations of the cloud fractions on the in-scattering side and large overestimations on the shadowing side. Realistic cloud scenes include shadowing and in-scattering simultaneously and it should be investigated, to which extent these two effects cancel.

The methodology could be applied to the satellite observations, e.g. HARP2 data. The cloud fraction retrieval should be compared to the obtained pixel-by-pixel cloud fraction of images captured by the OCI instrument onboard PACE, which operates at a higher spatial resolution of $1.2 \times 1.2 \, \mathrm{km}^2$. When applied to satellite observations, it also needs to be investigated for which regional coverage measurements at scattering angles in cloudbow and glint can be delivered nearly simultaneously. The retrieved cloud fraction corresponds to that of the pixel observing the sun-glint. Since the viewing direction of the corresponding pixels is not constant, the retrieved cloud fraction will be biased compared to the vertically projected cloud fraction which is commonly used as model diagnostic. This bias will vary regionally and seasonally. A detailed investigation on the distribution of this bias should be performed and based on this a bias correction method needs to be developed.

The derived cloud fraction of shallow cumulus clouds over ocean should be compared to the results by Dutta et al. (2020), who found cloud fraction reductions of more than 0.4 when the spatial resolution error is corrected. For climate and weather model validation, the global cloud fraction is an important quantity. The method we have presented provides the cloud fraction only for liquid water clouds over the ocean. Therefore our method needs to be adapted for ice clouds, which should be relatively straightforward by replacing the degree of polarization in the cloudbow by the intensity at a scattering angle outside the sun-glint region. The development of a cloud fraction retrieval over land surfaces is more challenging because land surface reflection causes only weak polarization. However, the Rayleigh scattering in clear regions between the clouds produces a strong polarization signal that should contain information about the cloud fraction and could be used similarly to the glint polarization in the retrieval method presented here.

*Author contributions.* CE performed the sensitivity studies, set up the retrieval method, performed the tests on specMACS observations, and wrote the manuscript. VP provided the specMACS data, including the observed polarized radiances as well as the retrieval results (cloud fraction and cloud microphysics). This work has been inspired by MMs work on cloud fraction retrieval for exoplanets using polarized observations. MM and BM helped to set up and improve the retrieval methodology. All authors contributed to the interpretation of the results and helped to improve the manuscript.

*Competing interests.* At least one of the (co-)authors is a member of the editorial board of Atmospheric Measurement Techniques.

*Acknowledgements.* We thank the two anonymous reviewers for valuable comments that helped to improve the manuscript.

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
