# Peer review of "Retrieval of cloud fraction and optical thickness of liquid water clouds over ocean from multi-angle polarization observations"

_EGUsphere, 2024_

## Referee Comment (RC2)

**General comments:**

This manuscript presents a proof of concept for a method that can be used to retrieve cloud optical thickness and cloud fraction from multi-angle polarization observations. In particular from measurements at two viewing angles: one within the cloudbow at a scattering angle of approximately 140° and a second in the sun-glint region or at a scattering angle of approximately 90°. Further, the method consists of a look-up table generated using a 1D radiative transfer code. In addition, the authors provide information on the theoretical basis of their approach based on a limited (but sufficient) sensitivity analysis under idealized setups using the same 1D radiative transfer code.

Overall, the manuscript well suited for AMT and I believe the community can benefit from it. Also, the methods used seem to be robust and the language is fluent and precise. However, there some aspects of the manuscript that I am concerned about (see my comments below), and I believe are required to be addressed and clarified.

**Specific comments:**

1. The authors claim that they are presenting a retrieval algorithm that can be used to retrieve cloud optical thickness and cloud fraction from multi-angle polarization observations. Despite the authors' claim, the contents of the manuscript present the theory and proof of concept that the theory would work. Let me further explain this: an algorithm is defined as "a process or set of rules to be followed in calculations or other problem-solving operations". Here is an example to put things in perspective: you can have multiple lookup tables for a wide range of surface and aerosol characteristics, plus different cloud types. For this method to be called an algorithm, it must contain an automated mechanism to select between those lookup-tables. Or, for example, it should also contain automated steps for discriminating the cloud phase. I believe this issue can be addressed by either rephrasing the text and correcting the statement, or by developing the further steps required and expanding the simulations for the method to be called an algorithm. Also, the entire manuscript, including the title and abstract should reflect this matter.

2. Under real conditions, the measurements will contain some degrees of error associated with them. It is not clear how the instrumental noises are accounted for in the tests provided in this manuscript.

3. Given the fact that this manuscript is based on very few test scenes and idealized simulations, I find a bit difficult to understand why addressing the effects of the 3D radiative transfer should be a separate manuscript.

4. The analysis performed to evaluate the sensitivity of the responses to aerosols needs to be expanded to address as the total AOD is probably not the only affecting factor. In particular, the above-cloud-AOD, aerosol composition and size can also be important.

5. It is not clear whether the approach is intended to be developed for ice or liquid clouds or both. Also, whether for application over land or water. This has to be clarified along the manuscript including the title and abstract.

6. The authors claim that the approach is suitable for space-born multi-angle polarimetric remote sensing and they are waiting for the PACE data to become available. But it could also be test on the PARASOL-POLDER measurements (https://www.aeris-data.fr/catalogue/?checkBoxCriteria=%7B%22projects%22%3A%5B%22SPATIAL.PARASOL%22%2C%22SPATIAL.POLDER%22%5D%7D#masthead) as this data is available?

7. Adding some thoughts on how the authors are planning to validate the retrieved cloud optical thickness could be nice.

8. I believe, a viewing angle with a scatting angle that is exactly 90 and 140 degrees may not always be available. What is the protocol for such cases? Also, I see that the authors are using the word "approximately" in this context. It would be nice to add some words on the impacts of this non-exact scattering angles on the accuracy of the retrievals.

9. Some words on how the retrieval accuracy can be affected by using only the central band wavelengths and not applying the instrument response function can be beneficial.

**Technical corrections:**
1. In my opinion, the abstract does not well represent the contents of the manuscript (i.e., data used, theoretical analysis presented, testing performed and applications).
2. The story line in the Introduction section seems a bit off (sounds to me that some random information is being given to reader to fill the space) and can benefit from some revision. Here are some examples:
   a. The text given between lines 23-25 seems redundant or at least requires rephrasing. Also, I don't get how as the authors state the need for more observations is linked to algorithm development (that is the further exploitation of the existing data). Also, it does not connect well the previous and the next paragraphs.
   b. Something seems to be missing between the third and fourth paragraphs. And then it jumps from clouds to aerosols. And then jumps to clouds.
   c. Then the authors jump to giving information about the upcoming satellite missions.

3. It will be nice to have the hypothesis of the manuscript defined in the introduction and also specify why such a product is required / what this

product adds to the existing information.

4. The structure of section 2 is not very straight-forward and can benefit from some adjustments. Here are some suggestions:
    a. I would start the Methodology section with some brief information on what is it that is intended to be addressed and how (i.e., testing and explaining the theory behind the method, followed by the fact that synthetic measurements are produced …). Followed by two sub-sections that explain the methods for the theory and testing parts, respectively. Include all the methodology here, unless it is not possible.
    b. The sensitivity analysis performed seems like a result to me that explains the theory behind the approach. For this reason, I would create a new section 3 and dedicate it to the sensitivity analysis and call it something like "theoretical basis".
    c. The current section 3 can then be the new section 4.

5. It will be of a great importance to provide the bi-directional reflectance and polarization functions corresponding to the surfaces used.

6. Both "clear sky" and "cloud free" terms are used in the manuscript. Please harmonize unless the meaning is not different (in that case define it more clearly).

7. When talking about cloud layer height, is that cloud top height that you are referring to or base height?

8. Paragraphs 3 and 4 of section 4 contain plenty of unnecessary repetition of information that does not help concluding anything. Rephrase, shorten or remove.

9. Figure 5:
    a. It will be useful to have likes of Figure 5 for different land types somewhere in the manuscript (appendix would do as well).
    b. Write in the caption that the figure is for the case of ocean.
    c. AOD: is it AOD above cloud or total column AOD?
    d. It would make it easier to read the figure if you add just beside the figure (where a, b, c, … are) some small text that can be related to the parameter that is being evaluated. Example: a [tau]

10. Line 195: (tau_c), (c)
11. Line 196,197: are separated
12. Line 200: realize what?
13. Line 202: I believe what is important here is the cloud top height, not the base. Rephrase the sentence based on cloud top height.

14. Line 206-207: The sentence "P(110) …" does not seem to be based on the plot.
15. Info given in page 12 can go to the Methodology section explained above in comment 4.
16. Lines 201-202: explain the abbreviation the first time that the abbreviation is used. Also, do EUREC4A and HALO stand for something?
17. Line 236: the word "corresponding" seems strange there.
18. Lines 243-245: it is not very clear, please rephrase or expand.
19. Figure 6:
    a. Pannel names are difficult to see (font size is too small).
    b. The terms left and right are not very clear, maybe they could more clearly be labeled.
    c. Make the boarders of the shaded area more visible. It is hard to see it.
20. Line 255: total column AOD=0.08 or above-cloud-AOD?
21. Lines 268-269: omit "for this scene", and the second "which".
22. Line 287: Please further explain.
23. Line 282: comma is missing: palne, making
24. Line 298: comma is missing: ,for instance,
25. Line 331: MYSTIC?

---

## Author Comment (AC1)

**Response to reviewer #1**

We would like to thank the referee for the valuable comments on our manuscript, in particular also for providing many relevant references. Please find below the answers. We report in blue the comments by the reviewer, and our answers can be found below each comment in black.

**General Comments:**

This work presents a remote sensing technique that utilizes multi-angle polarization information to retrieve cloud fraction and cloud optical depth. Numerical experiments using 1D radiative transfer are presented to demonstrate the technique along with an extremely welcome field validation. The paper is well suited to publication in AMT. The technique is an important contribution as it provides a means to mitigate the weakness of coarse resolution that upcoming space-borne multi-angle polarimeters will have to deal with.

There are, however, several aspects of the paper that should be expanded before publication to give a more complete accounting of the technique. These include a discussion of the applicability to ice phase clouds, the definition of cloud fraction, and the issues inherent to the application of the retrieval to clouds with 3D geometries. The 3D nature of the cloudy atmosphere is an inextricable component of multi-angle retrievals such as this one and should be addressed within this paper, even if only with a combination of qualitative and idealized quantitative arguments. I therefore recommend major revisions to this paper to address the specific comments listed below.

**Specific Comments:**

1. Exactly which of the instruments listed in the introduction is this technique applicable to? It seems to me that for general solar zenith angles, only hyperangular measurements will be able to provide the simultaneous scattering angles required, and HARP2 is focused on for what appears to be this reason. If this is the case then this should be made clear when the polarimeters are introduced in Lines 51 to 65. Will there be limitations in the regional coverage of this technique applied to HARP2 because of this due to scattering geometry requirements?

   Thank you for the comment. It is correct that the scattering angles should be provided nearly simultaneously, therefore HARP2 is ideally suited. The regional coverage needs to be investigated, but this is out of the scope of this publication. The applicability to SpexOne and 3MI also requires further investigation. We demonstrated for specMacs, which does not observe the two scattering angles simultaneously, that the cloud fraction retrieval still provides good results. We therefore believe that the method could also be applied to SpexOne and 3MI.

   We have included the following in the introduction to clarify that HARP2 is the ideal instrument for our method:

   "Since cloud structures change rapidly, observations at the two suggested scattering angles should ideally be taken nearly simultaneously. HARP2 is designed to provide

such observations, so it is ideally suited for our proposed method. For SPEXone and 3MI, further investigations on the collocation of the observations would be required."

And in the outlook we included the following statement on the regional coverage:

"When applied to satellite observations, it also needs to be investigated for which regional coverage measurements at scattering angles in cloudbow and glint can be delivered nearly simultaneously."

2. I also wonder whether the choice of wavelength is based on HARP2's hyperangular channel or whether there would be some other more optimal wavelength for this technique.

Yes, we have indeed chosen the wavelength of 667 nm, because HARP2 provides the multi-angle observations that are needed. We clarified this in the introduction.

We have now put the focus of the paper on cloud fraction retrieval over the ocean, for which 667 nm is well suited. For cloud fraction retrievals over land, shorter wavelengths are more appropriate, because the retrieval over land requires Rayleigh scattering between the clouds.

3. The technique is stated to retrieve cloud fraction. I am left wondering exactly how this cloud fraction is defined. The definition will tend to depend on the purpose of the product, which is not made clear. If the purpose is to retrieve the geophysical variable of cloud fraction for model evaluation, then that is typically defined based on the projected area of clouds onto a space oblique plane. On the other hand, if the point is to identify the presence of clouds within pixels in an image, then that is a completely different target for the retrieval when non-nadir views are considered, as they are here. The definition of the target/truth will determine how the output of the retrieval is validated and also guide how a potential product is used and so this should be made clear. Perhaps the goal is to produce a resolution-independent estimate of cloud fraction due to the potential to provide a continuous output? This would certainly be an important contribution as most moderate-resolution satellite cloud masking products are not necessarily designed to explicitly estimate cloud fraction but rather to mask pixels for a variety of purposes.

We present a method that does not depend on spatial resolution and does not require any threshold values for cloud detection. It yields a continuous cloud fraction estimate that could be used for climate and weather model validation. We clarified this in the abstract and in the introduction.

A clearer target and application for the retrieval may help to focus the introduction which, in its present form, is very close to just a listing of different retrieval methods and instruments. Ideally, this section should present a problem to which the proposed technique is a solution. For example, how bad is the problem of unresolved clouds at 2-4 km resolution vs. 1 km? Why can't we just tune threshold algorithms to retrieval pixel-by-pixel cloud fraction at coarse resolution? Just a few of many relevant references: (Stubenrauch et al., 2024; Dutta et al., 2020, Wielicki, B. A., and  L. Parker, 1992)

We thank the reviewer for these important references. In the introduction, we included an outline on the problems related to the definitions of cloud and cloud fractions, in particular the dependence of cloud fraction on the spatial resolution of the observations when the cloud fraction is just defined as the ratio of cloudy pixels to total number of pixels in a given domain. When we derive the cloud fraction only from the reflected radiation at two angles, we overcome the problems related to spatial resolution of the observation.

If the choice of target is a pixel-by-pixel cloud fraction, then which pixel is it? Is it the pixel observing at 140 degrees or the one observing at 90 degrees? For general cloud geometries, observations at these two scattering angles will not observe the same fractional coverage of their field of view (and both will be distinct from the vertically projected cloud fraction). This may seem relatively minor relative to the precision of the technique (which is evaluated using specMACS) but will introduce an instability of the algorithm (i.e., a systematic error) to the target cloud type and the solar zenith angle which will alias into regional and seasonal variability and may be quantitatively significant. This is a fundamental and unavoidable aspect of the technique (as it is proposed here) and should be discussed especially as to how it will affect the use of the product to measure its target.

We retrieve the cloud fraction for the pixel observing the sun-glint. We found in our sensitivity studies that P(140°) over the ocean is almost insensitive to cloud fraction (see left panels in Fig. 6, only for very small optical thicknesses, P(140°) depends on cloud fraction). For shallow cumulus clouds, which are the focus of this paper, this difference is not extremely large.

In section 3.3 we included the following sentence for clarification:

"Note that, since the cloud fraction is derived from the observation in the sun-glint, the retrieved cloud fraction corresponds to the cloud fraction of the pixel observing the glint."

4. The same issues apply to the retrieval of cloud optical depth. For a model, optical depth is precisely defined as a vertical integral. Exactly how is this retrieved quantity defined here?

Vertical optical thickness (definition as in RT) shall be retrieved, and this is input to radiative transfer simulations to produce the lookup table. Since the lookup-table is produced for the specific sun-observation geometry, the slant path through the cloud layer is correctly taken into account.

Of course there are other systematic errors: as we show in the additional 3D simulations, the retrieved optical thickness is generally too small, which is due to the fact that 3D clouds are less reflective than 1D cloud layers. In 3D, photons are scattered through the cloud sides towards the surface. This problem is well-known in all cloud optical thickness retrieval algorithms.

5. Due to the required choice of scattering geometry the observation at 90 degrees or 110 degrees will tend to observe regions of the surface that are shadowed by the

cloud (i.e. when compared to near-backscatter). The technique seems to rely on observations of polarized surface reflection to determine cloud fraction. The shadowed regions of the surface will not provide this strong polarization signal despite being clear. It seems that this will induce a systematic error in the technique that varies with the shadow fraction of the fields of view (which will also differ between the two views). Shadow fraction will vary with cloud geometry such as area, spacing and cloud-base height and aspect ratio. While I appreciate and support the stated intention to examine the retrieval using 3D radiative transfer with complex cloud fields derived from Large Eddy Simulations in a subsequent study, this does not preclude the need to present and explain these basic features of the retrieval within this paper, perhaps with simple idealized clouds such as cuboids. It would also be interesting to see whether there is any detectable signals of these effects in the specMACS data, though this would require development of a pixel-by-pixel shadow mask.

We included a new section with 3D simulations to demonstrate the basic 3D effects (shadowing and in-scattering) impacting the retrieved cloud fraction in opposite directions.

Also we studied randomly generated simple shallow cloud fields, for which the cloud fraction is overestimated due to cloud shadows.

We agree that it would be interesting to look for the cloud shadow effects in specMacs data, but this is out of the scope of this paper.

6.  The authors state on Line 218 that the technique will deliver accurate cloud fraction and cloud optical thickness over the ocean. This statement seems overly strong in the case of cloud optical depth even in the highly simplified case of plane-parallel atmospheres. The estimation of the cloud optical depth from the cloud-bow degree of polarization will have a correlated error with the determination of the droplet effective radius and droplet effective variance. For example, the magnitude scaling parameter in the least squares fit for the shape of the cloudbow polarization pattern is implicitly sensitive to both cloud fraction and optical depth. It seems that to properly understand the error characteristics, all four parameters should be jointly retrieved.

Yes, we agree. In the cloud retrieval algorithm for e.g. HARP2, as many parameters as possible should be jointly retrieved. We investigated how much the retrieval results for specMacs change when we combine the retrieval with the cloud microphysics retrieval from the cloudbow and we find significant impacts on the optical thickness retrieval as to be expected from the sensitivity study shown in Fig. 6. The retrieval results for the selected scenes are listed in Table 1, for the retrieval assuming constant reff and veff values and for the retrieval using reff and veff from the cloudbow retrieval. The impact on the retrieved cloud fraction is relatively small.

7.  There is no mention of ice within this paper. Some discussion of whether this technique works for ice phase clouds or mixed phase clouds should be included. Due to the coarse resolution, there will also be a further lack of uniqueness in the cloud phase in the actual data when compared to moderate resolution (1 km) imagers. This possible limitation should be discussed but even if the technique relies on liquid

scattering signals, this technique will still be a valuable contribution because most small clouds (and hence partially cloudy pixels) originate from low-level liquid clouds.

Ice clouds do not produce a cloudbow, so this method is only applicable for liquid water clouds. This is now discussed in the introduction:

"In order to obtain the global cloud cover, the method needs to be extended to ice clouds and to observations over land surfaces. Since ice clouds do not produce a cloudbow, the methodology can not directly be applied. However, it would be possible to replace the degree of linear polarization in the cloudbow region by an intensity observation at the same angle to retrieve the ice cloud optical thickness. …"

The discussion of non-oceanic surfaces appears slightly incomplete. The authors mention that for brighter unpolarized surfaces the technique is impossible but do not come to a conclusive statement about dark unpolarized surfaces, simply stating that the retrieval will be more uncertain. For the case with the unpolarized surface, the retrieval is reliant on the molecular polarization signal and there will be an ambiguity about whether the polarization signal is due to cloud top height or cloud fraction. This sensitivity to cloud top height is not examined for a dark unpolarized surface. It would be helpful to reference the accuracy with which the DoLP can be measured so that the relative statements about uncertainty are translated to actual retrieval uncertainties.

Yes, we agree. For this paper we now limited the methodology for liquid clouds over the ocean. For land, shorter wavelengths would be advantageous to have more Rayleigh polarization caused by molecular scattering between the clouds and to have less signal from the surface. But for sure this method would become far more complicated, because trace gas and aerosol vertical profiles will matter.

8. I am also curious as to whether the retrieval concept presented is overly simplified and doesn't fully exploit the observational information content. There are some dependences of the retrieval on wind speed and aerosol optical depth documented within the paper. For the purposes of reducing systematic errors and ensuring proper uncertainty propagation, the more variables that are explicitly retrieved (even if with strong priors), the better. Given the hyperangular observations from HARP2 and other multi-spectral observations, is it not possible to jointly retrieve wind speed/surface roughness and aerosol optical depth (e.g., Knobelspiesse et al., 2021) in combination with cloud fraction using this information and some prior information on aerosol composition? I think it would be valuable for the authors to discuss the feasibility of this as a possible extension to their work.

Yes, we agree. This method should be combined with the overall retrieval methodology. Ideally AOD and wind speed should be retrieved simultaneously. The operational algorithms for PACE are of course more advanced, and rather than extending our method, we would suggest that the PACE team adds the cloud fraction retrieval as an addition to their methodology.

The following text has been added to clarify that it is advantageous to combine the retrieval with other algorithms: "For the selected scenes, we generated lookup-tables

using the cloud size distribution parameters from the cloudbow retrieval in addition. Of course, it would make sense to combine the method with further retrieval algorithms, e.g., with the simultaneous aerosol and ocean glint retrieval by \citet{knobelspiesse2011}. When accurate a priori information about wind speed and aeorosol optical thickness is included, one should also take into account the filter function of the instrument rather than running monochromatic simulations to generate the lookup-table. These improvements are not necessarily needed to demonstrate the method for a few specific cases, which is the purpose of this study."

9. The validation of the technique against field data is an extremely valuable component of the paper. Some greater discussion of the threshold-based cloud mask that the technique is being compared against is warranted. The features used in the cloud mask are listed but the tuning of the thresholds is not. Is the cloud mask clear-conservative or cloud conservative? Or is it designed to optimally estimate cloud fraction over 2.5 km regions? This is important for understanding any agreement or lack thereof between the coarse resolution technique proposed here and the high-resolution threshold-based technique. As I understand it, the high-resolution reference mask also makes use of polarization features to separate sunglint and from non-sunglint. Perhaps this point, and the ambiguity of masking in sunglint regions with just intensity measurements should be more emphasized as a strength of the technique

We have added a paragraph describing the cloud detection algorithm:

"We also employed the cloud detection method outlined in \citet{poertge2023} on the images of the selected scenes to determine the geometrical cloud fraction of the scenes at the original (high) resolution of the images. The method is based on the algorithm described in \citet{otsu_1979} which determines a threshold to seperate the pixels of an image in two classes (here: cloudy versus non-cloudy) based on a brightness histogram. Such threshold based cloud detection algorithms often struggle with the bright sun-glint reflection. Our algorithm uses the parallel component of polarized light in which the reflectance of the sun-glint is reduced which in turn reduces the number of incorrect classifications in (cloud free) sun-glint areas. The algorithm distinguishes between cloud-free, low to medium cloud coverage and high cloud coverage. For cloud-free scenes it uses the data of the blue channel, for scenes with low/medium cloud coverage the data of the red channel, and otherwise the normalized red ($r$) to blue ($b$) ratio ($\text{nrbr} = (b-r)/(b+r)$). This procedure was found by tuning the cloud mask for many different scenes (both over land and over water)."

10. For the specMACS data, I would appreciate seeing retrievals that used the cloudbow reff/veff as input as this is the best guess for the retrieval and would reduce any possible error compensation in the validation.

We agree that the reff/veff from the cloudbow retrievals should be used as input to the cloud fraction and optical thickness retrieval. It is obvious from Fig.6 that the effective radius impacts the polarization at 140° scattering angle because the angular

pattern of the scattering phase matrix, in particular in the cloudbow region, depends on the effective radius. However, the intention of our paper is to demonstrate the methodology and for this purpose it is nice to plot all observational points in one lookup table. For the discussed scenes we have now computed lookup-tables using the correct microphysics as input, these are shown in the following Figure (not included in the paper):

[Figure]

The title of the figure includes the effective radius and the effective variance obtained from the cloudbow retrieval and the retrieved optical thickness (tau) and cloud fraction (cf).

In the paper we have included two additional columns in Table 1 including the retrieved optical thickness and the cloud fraction using the microphysics retrieved from the cloudbow. As expected, we find the largest deviation for the scene where the effective radius deviates most from 10 µm (s6).

**Technical Comments:**

Line 93: I was a little confused on a first read by the statement that all simulations had a liquid cloud layer located from 2-3 km and same droplet effective radius. It would be helpful to note that this restricted setup is just to illustrate the main sensitivity to cloud fraction and optical depth and that more sensitivity tests are performed later.

ok. Included a note here.

Line 304: Sun-glint

ok.

**Relevant References:**

Wielicki, B. A., and  L. Parker (1992),  On the determination of cloud cover from satellite sensors: The effect of sensor spatial resolution, *J. Geophys. Res.*,  97(D12),  12799–12823, doi:10.1029/92JD01061.

Stubenrauch, C.J., Kinne, S., Mandorli, G. *et al.* Lessons Learned from the Updated GEWEX Cloud Assessment Database. *Surv Geophys* (2024). https://doi.org/10.1007/s10712-024-09824-0

Dutta, S.,  Di Girolamo, L.,  Dey, S.,  Zhan, Y.,  Moroney, C. M., &  Zhao, G. (2020).  The reduction in near-global cloud cover after correcting for biases caused by finite resolution measurements. *Geophysical Research Letters*,  47, e2020GL090313. https://doi.org/10.1029/2020GL090313

Knobelspiesse, K., Ibrahim, A., Franz, B., Bailey, S., Levy, R., Ahmad, Z., Gales, J., Gao, M., Garay, M., Anderson, S., and Kalashnikova, O.: Analysis of simultaneous aerosol and ocean glint retrieval using multi-angle observations, Atmos. Meas. Tech., 14, 3233–3252, https://doi.org/10.5194/amt-14-3233-2021, 2021.

---

## Author Comment (AC2)

**Response to Reviewer #2:**

We would like to thank the referee for the valuable comments on our manuscript. Please find below the answers. We report in blue the comments by the reviewer, and our answers can be found below each comment in black.

**General comments:**

This manuscript presents a proof of concept for a method that can be used to retrieve cloud optical thickness and cloud fraction from multi-angle polarization observations. In particular from measurements at two viewing angles: one within the cloudbow at a scattering angle of approximately 140° and a second in the sun-glint region or at a scattering angle of approximately 90°. Further, the method consists of a look-up table generated using a 1D radiative transfer code. In addition, the authors provide information on the theoretical basis of their approach based on a limited (but sufficient) sensitivity analysis under idealized setups using the same 1D radiative transfer code.
Overall, the manuscript well suited for AMT and I believe the community can benefit from it. Also, the methods used seem to be robust and the language is fluent and precise. However, there some aspects of the manuscript that I am concerned about (see my comments below), and I believe are required to be addressed and clarified.

**Specific comments:**

1. The authors claim that they are presenting a retrieval algorithm that can be used to retrieve cloud optical thickness and cloud fraction from multi-angle polarization observations. Despite the authors' claim, the contents of the manuscript present the theory and proof of concept that the theory would work. Let me further explain this: an algorithm is defined as "a process or set of rules to be followed in calculations or other problem-solving operations". Here is an example to put things in perspective: you can have multiple lookup tables for a wide range of surface and aerosol characteristics, plus different cloud types. For this method to be called an algorithm, it must contain an automated mechanism to select between those lookup-tables. Or, for example, it should also contain automated steps for discriminating the cloud phase. I believe this issue can be addressed by either rephrasing the text and correcting the statement, or by developing the further steps required and expanding the simulations for the method to be called an algorithm. Also, the entire manuscript, including the title and abstract should reflect this matter.

OK. We fully understand this point. Indeed, our intention is to present a new idea to retrieve cloud fraction from satellite observations and to provide a proof-of-concept. We have renamed "algorithm" to "method" throughout the paper.

2. Under real conditions, the measurements will contain some degrees of error associated with them. It is not clear how the instrumental noises are accounted for in the tests provided in this manuscript.

At this point, we assume that the measurement errors are smaller than errors due to uncertain assumptions (e.g. about wind speed and aerosol optical properties). In a complete "algorithm" this certainly needs to be included, but for the proof of concept we believe that this is not necessary.
As a demonstration we apply the method on real airborne observations (Section 5), and we obtain consistent results from two methods to derive the cloud fraction, which should be sufficient for demonstration purposes.

3. Given the fact that this manuscript is based on very few test scenes and idealized simulations, I find a bit difficult to understand why addressing the effects of the 3D radiative transfer should be a separate manuscript.

We included a new section with 3D radiative transfer simulations to demonstrate the basic 3D effects (shadowing and in-scattering) impacting the retrieved cloud fraction in opposite directions.

Also we studied randomly generated simple shallow cloud fields, for which the cloud fraction is overestimated due to cloud shadows.

A more detailed study including realistic LES clouds should be presented in a separate manuscript.

4. The analysis performed to evaluate the sensitivity of the responses to aerosols needs to be expanded to address as the total AOD is probably not the only affecting factor. In particular, the above-cloud-AOD, aerosol composition and size can also be important.

We performed additional sensitivity analysis for different aerosol types, different aerosol layer heights. and different size distributions (shown in the figures below). We find significant impacts on P(140), and only very small impacts on P(110) and P(90), where the degree of polarization is dominated by the sun glint. Therefore, for the cloud fraction retrieval, one needs to take into account only the total aerosol optical thickness. Other parameters, such as aerosol size distribution, composition and vertical profile have a smaller influence and can safely be neglected. For the optical thickness retrieval however, detailed aerosol characterization is more important.

[Figure]

5. It is not clear whether the approach is intended to be developed for ice or
liquid clouds or both. Also, whether for application over land or water. This
has to be clarified along the manuscript including the title and abstract.

For this paper, we have now limited the method for liquid clouds over the ocean (the title has
been changed accordingly). We added the following in the introduction for clarification:

"The global cloud cover cannot be obtained using our method because it is developed only for
liquid clouds over ocean.
Since ice clouds do not produce a cloudbow, the methodology cannot directly be applied to
determine the cloud fraction and the optical thickness of ice clouds. However, it would be
possible to replace the degree of linear polarization in the cloudbow region by an intensity
observation at the same angle to retrieve the ice cloud optical thickness. For retrievals over
land, polarization due to surface reflection is too small to be used for a cloud fraction retrieval.
Therefore an additional method needs to be developed which could use the strong polarization
caused by Rayleigh scattering between the clouds to obtain information about the cloud fraction.
This method should use shorter wavelengths which are mostly insensitive to surface properties
and for which the Rayleigh scattering contribution is much higher.

6. The authors claim that the approach is suitable for space-born multi-angle
polarimetric remote sensing and they are waiting for the PACE data to
become available. But it could also be test on the PARASOL-POLDER
measurements (https://www.aeris-
data.fr/catalogue/?checkBoxCriteria=%7B%22projects%22%3A%5B%22SPA
TIAL.PARASOL%22%2C%22SPATIAL.POLDER%22%5D%7D#masthead) as
this data is available?

Yes, it would be possible to also test it on POLDER measurements, but this is also a separate
study.

7. Adding some thoughts on how the authors are planning to validate the
retrieved cloud optical thickness could be nice.

We have now included the 3D radiative transfer simulations to generate synthetic observations
with known input. Using this, we also validated the optical thickness retrieval and found that it is
generally underestimated for cases where 3D cloud scattering plays a role.

8. I believe, a viewing angle with a scattering angle that is exactly 90 and 140
degrees may not always be available. What is the protocol for such cases?
Also, I see that the authors are using the word "approximately" in this context.
It would be nice to add some words on the impacts of this non-exact
scattering angles on the accuracy of the retrievals.

For specMACS we could not get 90° and 140° simultaneously, therefore we have used 110° and 140° and we show that this works equally well. It is only important to have one scattering in the cloudbow and another one in the sun-glint region, they do not need to be exactly at 90° and 140°. This has also been clarified in the text (Section 6):

"We also generated lookup tables using scattering angles of 140° and 110° and again obtained a well-separated lookup-table grid. This demonstrates that it is not important to choose exact scattering angles to set up the retrieval, one only has to make sure that one angle includes the cloudbow and the second angle a part of the sun-glint."

9. Some words on how the retrieval accuracy can be affected by using only the central band wavelengths and not applying the instrument response function can be beneficial.

We added the following to the description of the setup of the lookup tables for specMACS: "For the selected scenes, we generated lookup-tables using the cloud size distribution parameters from the cloudbow retrieval in addition. Of course, it would make sense to combine the method with further retrieval algorithms, e.g., with the simultaneous aerosol and ocean glint retrieval by \citet{knobelspiesse2011}. When accurate a priori information about wind speed and aeorosol optical thickness is included, one should also take into account the filter function of the instrument rather than running monochromatic simulations to generate the lookup-table. These improvements are not necessarily needed to demonstrate the method for a few specific cases, which is the purpose of this study."

**Technical corrections:**

1. In my opinion, the abstract does not well represent the contents of the manuscript (i.e., data used, theoretical analysis presented, testing performed and applications).

We have revised the abstract accordingly.

2. The story line in the Introduction section seems a bit off (sounds to me that some random information is being given to reader to fill the space) and can benefit from some revision. Here are some examples:

a. The text given between lines 23-25 seems redundant or at least requires rephrasing. Also, I don't get how as the authors state the need for more observations is linked to algorithm development (that is the further exploitation of the existing data). Also, it does not connect well the previous and the next paragraphs.
b. Something seems to be missing between the third and fourth paragraphs. And then it jumps from clouds to aerosols. And then

c. Then the authors jump to giving information about the upcoming
satellite missions.

3. It will be nice to have the hypothesis of the manuscript defined in the
introduction and also specify why such a product is required / what this
product adds to the existing information.

We have also revised the introduction. We have now put the focus on the cloud fraction, on
problems related to its definition and on the problems related to the observation of the cloud
fraction from space. In particular it depends on spatial resolution for most cloud fraction retrieval
algorithms and also on settings of various thresholds. Due to the limited spatial resolution, cloud
fraction is often overestimated. Our method provides a continuous and resolution-independent
estimate of the cloud fraction, and we demonstrate using specMACS observations that this
correlates well to the cloud fraction derived from observations at very high spatial resolution of
10m.

4. The structure of section 2 is not very straight-forward and can benefit from
some adjustments. Here are some suggestions:
a. I would start the Methodology section with some brief information on
what is it that is intended to be addressed and how (i.e., testing and
explaining the theory behind the method, followed by the fact that
synthetic measurements are produced …). Followed by two sub-
sections that explain the methods for the theory and testing parts,
respectively. Include all the methodology here, unless it is not
possible.
b. The sensitivity analysis performed seems like a result to me that
explains the theory behind the approach. For this reason, I would
create a new section 3 and dedicate it to the sensitivity analysis and
call it something like "theoretical basis".
c. The current section 3 can then be the new section 4.

Thank you for your suggestion, we have revised the structure accordingly. The "Methodology"
Section 2  includes the description of the model setup for 1D and 3D simulations. Section 2
includes the results of the 1D simulations and the setup of the retrieval lookup table. Section 4
includes the results of the 3D simulations and the investigation of the impact of 3D scattering on
the retrieval accuracy (retrieval test based on synthetic observations). Section 5 includes the
retrieval test on real airborne observations.

5. It will be of a great importance to provide the bi-directional reflectance and
polarization functions corresponding to the surfaces used.

This information was given in the results section: "we include an ocean surface which is
modelled using the reflectance matrix based on the Fresnel equations convolved with a

Gaussian kernel to account for the ocean waves (Mishchenko and Travis, 1997; Tsang et al., 1985; Cox and Munk, 1954a, b)." We moved this now to the methodology section. For land surfaces we only included Lambertian surfaces.

6. Both "clear sky" and "cloud free" terms are used in the manuscript. Please harmonize unless the meaning is not different (in that case define it more clearly).

Ok, now we consistently use "clear sky".

7. When talking about cloud layer height, is that cloud top height that you are referring to or base height?

We always refer to "cloud top height", this is now included in the text.

8. Paragraphs 3 and 4 of section 4 contain plenty of unnecessary repetition of information that does not help concluding anything. Rephrase, shorten or remove.

OK. The summary has been shortened.

9. Figure 5:
a. It will be useful to have likes of Figure 5 for different land types somewhere in the manuscript (appendix would do as well).

As mentioned, we now focus on retrieval over the ocean. Over land, it becomes far more complicated because cloud height will become important as well as the vertical profile of the aerosols.

b. Write in the caption that the figure is for the case of ocean.
OK.

c. AOD: is it AOD above cloud or total column AOD?
It is the total column AOD.

d. It would make it easier to read the figure if you add just beside the figure (where a, b, c, … are) some small text that can be related to the parameter that is being evaluated. Example: a [tau]
The label of the x-axis clearly shows the parameter that is evaluated.

10. Line 195: ($tau_c$), (c)
Not sure what you mean here. Comma after $tau_c$?

11. Line 196,197: are separated

OK.

Rephrased: "Therefore, is makes sense to combine the retrieval with an effective radius retrieval based on the cloudbow signature."

Corrected.

Thank you, we mean here the absolute value of P(110°). This has been corrected.

We prefer to leave the information on the observational data in Section 5 to keep the test on real observations separate from the theoretical part including synthetic observations. The definition of the signed degree of linear polarization has been moved to the "Methodology".

OK. Explained specMACS, EUREC4A and HALO where they are first used.

Replaced "corresponding" by "equivalent".

The reader is referred to the references Kölling et al. 2019 and Pörtge et al. 2023 for more details.

The panel names are now on a white background to see them more clearly.

We included the scattering angles as additional labels at the top of the images.

Done.

Clarified.

21. Lines 268-269: omit "for this scene", and the second "which".
This part has been rewritten, because we exchanged some of the scenes.

22. Line 287: Please further explain.
Rephrased.

23. Line 282: comma is missing: palne, making
Added comma.

24. Line 298: comma is missing: ,for instance,
Included.

25. Line 331: MYSTIC?
Included expansion of MYSTIC where it is first used.

---

## Referee Report (RR1)

Review of Egusphere-2024-1180:

The authors present a novel method for retrieving cloud fraction from multi-angle polarized measurements that observe both cloud-bow and oceanic sun-glint. The novel contribution of this work is especially in the utilization of the polarized sun-glint signal to retrieve a resolution-independent measure of cloud fraction. The manuscript is well-written and acts as a demonstration of the key physical dependencies of the retrieval, using a combination of 1D RT and simple 3D RT simulations. This work can form the basis for the potential development and more widespread validation of an operational algorithm.
For this version, I have three comments for which I recommend minor revisions to the manuscript.

First, the method relies on the presence of unambiguous liquid within the field of view. This becomes increasingly unlikely when combining multi-angle observations at coarse-resolution as is done here. Naturally, the final effect of undetected ice will depend on the details of the cloud-top phase algorithm that is utilized. I don't think that a detailed emulation of the role of the cloud-top phase algorithm in filtering errors is necessary for this work. However, no phase determination algorithm will be perfect. So, this study should also perform a similar sensitivity study using 1D RT on the effect of the presence of overlying optically thin ice layers (e.g., cirrus) on the retrieval.

Second, the utility of this product for testing scientific hypotheses relies on its algorithmic stability. While threshold-based, pixel-by-pixel cloud masks provide an unstable estimate of the cloud fraction when the cloud size distribution changes, they can be designed to be invariant to solar-viewing geometry. On the other hand, the method developed here depends on the cloud fraction in the sun-glint view, which will change in viewing zenith angle with solar zenith angle. The cloud fraction at an oblique view is not the same as the vertically projected cloud fraction that is used as a model diagnostic, for example. For a polar-orbiting satellite this difference will introduce a regionally and seasonally varying bias in the apparent vertically projected cloud fraction. The magnitude of this bias is one of the foremost controls on the utility of this proposed method. This concept should be discussed within the text, with a recommendation that this effect should be evaluated in further studies.

It appears the 3D RT simulations are at a single SZA of 50 degrees, which is associated with an overestimation of cloud fraction. The magnitude of the SZA-dependence of the retrieval will depend strongly on the realism of the cloud geometry. The 3D RT simulations utilize relatively large horizontal-to-vertical aspect ratios (5:1), when compared with typical cumulus, which might be more on the order of 2:1 or 1:1. Therefore, I don't see it as strictly necessary to quantify this with the simplified cloud fields presented here (though I would welcome it if the authors choose to also include 3D RT results for SZA=15 to contrast). However, it is very important to describe the concept so that readers can understand all of the expected behavior of the technique.

Third, I believe the value of the specMACS analysis is to validate the cloud fraction derived from the coarse-resolution polarization method. However, at the moment, there is just an intercomparison of two cloud masks. Based on figure 16, the pixel-by-pixel cloud mask is missing a lot of cloud, which makes its use as a reference for assessing cloud fraction not as helpful. As stated in the text, the pixel-by-pixel cloud mask is designed to identify candidates for microphysical retrievals, leading to a tendency to be cloud-conservative, rather than being designed to optimally estimate cloud fraction. Fig. 18 may therefore be underestimating the presence of a more significant underestimate by the new cloud fraction estimate.

Given the relatively small number of data-points selected for the comparison, there is not really a need to compromise accuracy by utilizing an automatic Otsu thresholding method. I recommend that the thresholds be tuned by hand to each scene to ensure that all cloud is included. This way ambiguities such as mentioned in Line 441-442 can be avoided.

---

## Referee Report (RR2)

Review Report for the manuscript entitled "**Retrieval of cloud fraction and optical thickness of liquid water clouds over ocean from multi-angle polarization observations**" with the identification number "**EGUSPHERE-2024-1180**".

**General comments:**

The manuscript has improved substantially, and my major concerns have been addressed properly by authors. Nevertheless, I have spotted a few points that I believe require to be attended before publication. For this reason, I suggest "**minor revisions**" for the manuscript at this stage. I have referred to these points in my comments below.

**Technical comments:**

1. Line 14 (Abstract): Please specify what kind of algorithms (e.g., ... existing "aerosol and cloud" retrieval algorithms).
2. Lines 52-64 (Introduction): This paragraph could benefit from some words on a recent paper published on the use of polarimetric measurements for retrieving sub-pixel cloud fraction: **Yuan, et al. 2024. Cloud detection from multi-angular polarimetric satellite measurements using a neural network ensemble approach, Atmos. Meas. Tech., 17, 2595–2610, https://doi.org/10.5194/amt-17-2595-2024.**
3. Line 58 (Introduction): The acronyms "GOME" and "SCIAMACHY" have been defined, but never used again. They can be removed.
4. Line 143 (Section 2.1): from? Do you mean "around"? Please clarify.
5. Line 144 (Section 2.2): Please specify the how the cloud optical thickness values are distributed between 1 and 50.
6. Line 265 (Section 4): "on" -> "to the".
7. Section 4.1 (optional): I believe combining figures 6 and 7 into one figure with two panels may help the reader to see the differences between both cases.
8. Figure 7 caption: Please write the full caption.
9. Figure 8 caption: Retrieval lookup table derived in section 3 "(Figure 2)" ... .
10. Figure 8 and 2: Please use color-codings that show a better contrast between values and variables (i.e., different shades of blue and black look very similar to the eye –at least to mine– and the contrast among different values of the same variable are not easy to spot). One could also try writing the values beside the corresponding lines and consider having two colors to discriminate the horizontal and vertical lines from each other. Also, if possible, try using two completely different symbols for the scattered dots (X and O, for example). One needs to focus a lot to see the different shapes. This applies to all the figures provided in the manuscript.
11. Lines 294-309 (Section 4.1): I sense that the paragraph could benefit from including a take-away message that concludes what conclusion can be derived from figure 8?
12. Figure 9: color coddings are not consistent with the other plots in the manuscript. Please harmonize. Is there a reason for why the scattered dots to be located on a horizontal line? Please mention why it is like that. Also, I don't fully understand what the linear regression lines in this figure tend to show/prove. Could you please elaborate on that?

13. Line 311 (Section 4.1): lines -> dots.
14. Lines 331-333 (Section 4.1): this part seems to be about the sun glint only. It would be beneficial to give similar explanations for the cloud bow as well.
15. Line 350 (Section 5): Is there a reference paper for the EUREC4A campaign? Seems like Stevens et al. 2021 is the one. If so, please include it here.
16. Figure 12: maybe worth it to say a few words on the difference among the points that fall exactly on the 1:1 line and those that fall apart from that.
17. Line 418 (Section 5.1): ... tau_1 and c_1 are, "respectively", ... .
18. Line 434 (Section 5.1): Do you mean that the "difference" in the cloud optical thickness is small?
19. Line 452 (Section 5.1): has -> have ; (i.e., smaller than 0.2).
20. General comment (Section 5): merge section 5 with subsection 5.1. As there is no subsection 5.2, I don't see a need for having 5 and then 5.1.
21. Section 6 (Conclusions and Outlook): Outlook -> outlook
22. Line 459 (Section 6): ... optical thickness of liquid clouds over ocean ...
23. Line 475 (Section 6): This demonstrates that ...

---

## Author Response (AR2)

**Answers to reviewer #1**

We thank the referee for constructive comments on the revised version of the manuscript. Please find below the answers. We report in blue the comments by the reviewer, and our answers can be found below each comment in black.

The authors present a novel method for retrieving cloud fraction from multi-angle polarized measurements that observe both cloud-bow and oceanic sun-glint. The novel contribution of this work is especially in the utilization of the polarized sun-glint signal to retrieve a resolution-independent measure of cloud fraction. The manuscript is well-written and acts as a demonstration of the key physical dependencies of the retrieval, using a combination of 1D RT and simple 3D RT simulations. This work can form the basis for the potential development and more widespread validation of an operational algorithm.
For this version, I have three comments for which I recommend minor revisions to the manuscript.

First, the method relies on the presence of unambiguous liquid within the field of view. This becomes increasingly unlikely when combining multi-angle observations at coarse-resolution as is done here. Naturally, the final effect of undetected ice will depend on the details of the cloud-top phase algorithm that is utilized. I don't think that a detailed emulation of the role of the cloud-top phase algorithm in filtering errors is necessary for this work. However, no phase determination algorithm will be perfect. So, this study should also perform a similar sensitivity study using 1D RT on the effect of the presence of overlying optically thin ice layers (e.g., cirrus) on the retrieval.

We added another scenario (h) to Fig. 4, where we included a sub-visible cirrus layer above the broken liquid water clouds. We find the following (included in text):
"In order to test how much the retrieval will be influenced by sub-visible cirrus clouds above the liquid clouds we added an optically thin cirrus layer with a cloud top height of 11 km and a geometrical thickness of 1 km (scenario (h)). The effective radius of the crystals is set to 30 μm and the parameterisation by Baum et al. 2014 is applied to obtain the ice cloud optical properties. Generally we see as expected a decrease in degree of polarization with increasing cirrus optical thickness at all scattering angles. This means that with increasing cirrus optical thickness the retrieved liquid water cloud optical thickness will slightly decrease and the retrieved cloud fraction will slightly increase. The sub-visible cirrus layer does not block the glint, therefore the retrieved cloud fraction corresponds approximately to that of the liquid clouds."

Second, the utility of this product for testing scientific hypotheses relies on its algorithmic stability. While threshold-based, pixel-by-pixel cloud masks provide an unstable estimate of the cloud fraction when the cloud size distribution changes, they can be designed to be invariant to solar-viewing geometry. On the other hand, the method developed here depends on the cloud fraction in the sun-glint view, which will change in viewing zenith angle with solar zenith angle.

The cloud fraction at an oblique view is not the same as the vertically projected cloud fraction that is used as a model diagnostic, for example. For a polar-orbiting satellite this difference will introduce a regionally and seasonally varying bias in the apparent vertically projected cloud fraction. The magnitude of this bias is one of the foremost controls on the utility of this proposed method. This concept should be discussed within the text, with a recommendation that this effect should be evaluated in further studies.

Included the following in the outlook section:
"The retrieved cloud fraction corresponds to that of the pixel observing the sun-glint. Since the viewing direction of the corresponding pixels is not constant, the retrieved cloud fraction will be biased compared to the vertically projected cloud fraction which is commonly used as model diagnostic. This bias will vary regionally and seasonally. A detailed investigation on the distribution of this bias should be performed and based on this a bias correction method needs to be developed. "

It appears the 3D RT simulations are at a single SZA of 50 degrees, which is associated with an overestimation of cloud fraction. The magnitude of the SZA-dependence of the retrieval will depend strongly on the realism of the cloud geometry. The 3D RT simulations utilize relatively large horizontal-to-vertical aspect ratios (5:1), when compared with typical cumulus, which might be more on the order of 2:1 or 1:1. Therefore, I don't see it as strictly necessary to quantify this with the simplified cloud fields presented here (though I would welcome it if the authors choose to also include 3D RT results for SZA=15 to contrast). However, it is very important to describe the concept so that readers can understand all of the expected behavior of the technique.

We need an observation in the cloudbow and a separate observation in the sun-glint. For SZA=15° the cloudbow and the sunglint overlap and the contributions of cloud and surface to the polarization signal are not well separated. In order to see the dependence on solar zenith angle, we have run the same setup with SZA=30° and SZA=70°.

For SZA=30°, we use a scattering angle of 110° for which we get a signal from the ocean glint, and we obtain very similar results as for SZA=50° for the particular shallow cumulus clouds:

[Figure]

And also for SZA=70° the method works as expected when we use a scattering angle of 60° for the cloud glint signal:

[Figure]

We find the behavior that is expected from the observation geometry: The overestimation of the cloud fraction increases with increasing solar zenith angle whereas the underestimation of cloud optical thickness decreases.

We also run a simulation for the broken cloud fields with an aspect ratio of 1:1, i.e. a cloud geometrical thickness of 0.5 km:

[Figure]

The overestimation of cloud fraction is increased compared to the aspect ratio of 5:1 and also the underestimation of cloud optical thickness is larger (as expected because more radiation can escape through the cloud sides).

Since the main topic of the paper is to present the idea of the cloud fraction retrieval and the paper is already quite long, we decided to not include these results. We have added the following general conclusion at the end of Section 4.2:

"All results presented in this section are for a solar zenith angle of 50°. We have run the same simulations for solar zenith angles of 30° and 70° and found a similar performance of the retrieval. Generally, with increasing solar zenith angle the overestimation of cloud fraction increases. The underestimation of cloud optical thickness on the other hand decreases with increasing solar zenith angle."

Third, I believe the value of the specMACS analysis is to validate the cloud fraction derived from the coarse-resolution polarization method. However, at the moment, there is just an intercomparison of two cloud masks. Based on figure 16, the pixel-by-pixel cloud mask is missing a lot of cloud, which makes its use as a reference for assessing cloud fraction not as helpful. As stated in the text, the pixel-by-pixel cloud mask is designed to identify candidates for microphysical retrievals, leading to a tendency to be cloud-conservative, rather than being designed to optimally estimate cloud fraction. Fig. 18 may therefore be underestimating the presence of a more significant underestimate by the new cloud fraction estimate.

Given the relatively small number of data-points selected for the comparison, there is not really a need to compromise accuracy by utilizing an automatic Otsu thresholding method. I

recommend that the thresholds be tuned by hand to each scene to ensure that all cloud is included. This way ambiguities such as mentioned in Line 441-442 can be avoided."

The goal of this intercomparison is to demonstrate the cloud fraction retrieval using the polarization of the glint. As we mention there are several reasons why they do not agree perfectly, which is probably always the case when comparing cloud fraction retrievals.

We think that using an automated approach is more reliable and comparable than using different thresholds for each scene to match our visual impression.
Note that the number of data points is not so small, in addition to the 6 "selected scenes" Figures 15 and 17 include the results of 70 further scenes which were automatically processed.

Also, we use this cloud detection algorithm to identify the pixels for which the cloud microphysics retrieval is performed. The output of this retrieval (reff, veff) is then used as a constraint for the cloud fraction/optical thickness retrieval (c1/tau1), so we would not like to introduce another inconsistency here.

For these reasons we have decided to keep the automatic cloud detection algorithm.

**Answers to reviewer #2**

We thank the referee for constructive comments on the revised version of the manuscript, in particular for carefully reading and commenting on the new section including 3D radiative transfer simulations. Please find below the answers. We report in blue the comments by the reviewer, and our answers can be found below each comment in black.

1. Line 14 (Abstract): Please specify what kind of algorithms (e.g., … existing "aerosol and cloud" retrieval algorithms).
Done.

2. Lines 52-64 (Introduction): This paragraph could benefit from some words on a recent paper published on the use of polarimetric measurements for retrieving sub-pixel cloud fraction: Yuan, et al. 2024. Cloud detection from multi-angular polarimetric satellite measurements using a neural network ensemble approach, Atmos. Meas. Tech., 17, 2595–2610, https://doi.org/10.5194/amt-17-2595-2024.

Thank you for this very interesting reference which we have added to the introduction as suggested.

3. Line 58 (Introduction): The acronyms "GOME" and "SCIAMACHY" have been defined, but never used again. They can be removed.
OK.

4. Line 143 (Section 2.1): from? Do you mean "around"? Please clarify.
Now included geometrical thickness of cloud layer and cloud top height.

5. Line 144 (Section 2.2): Please specify how the cloud optical thickness values are distributed between 1 and 50.
Added the specific values of the optical thickness values

6. Line 265 (Section 4): "on" -> "to the".
OK.

7. Section 4.1 (optional): I believe combining figures 6 and 7 into one figure with two panels may help the reader to see the differences between both cases.
Good point. We will make sure during the typesetting procedure that the two figures are on the same page, so that they can easily be compared.

8. Figure 7 caption: Please write the full caption.
OK.

9. Figure 8 caption: Retrieval lookup table derived in section 3 "(Figure 2)" … .
OK.

10. Figure 8 and 2: Please use color-codings that show a better contrast between values and variables (i.e., different shades of blue and black look very similar to the eye –at least to mine– and the contrast among different values of the same variable are not easy to spot). One could also try writing the values beside the corresponding lines and consider having two colors to discriminate the horizontal and vertical lines from each other. Also, if possible, try using two completely different symbols for the scattered dots (X and O, for example). One needs to focus a lot to see the different shapes. This applies to all the figures provided in the manuscript.
We included in most of the lookup table plots the values besides the lines and it should now be clear which lines correspond to which variable (since tau is in the range between 0 and 50 and cloud fraction in the range between 0 and 1). As suggested we are now using different symbols ('O' for in-scattering and 'X' for scattering).

11. Lines 294-309 (Section 4.1): I sense that the paragraph could benefit from including a take-away message that concludes what conclusion can be derived from figure 8?
We included a short conclusion and a transition to the next paragraph describing Fig.9 : "Fig. 8 shows that 3D scattering effects cause significant biases in the retrieval results for both the cloud fraction and the optical thickness. In order to validate these biases more quantitatively we show the retrieved data against the input data as scatter plots in Fig. 9."

12. Figure 9: color codings are not consistent with the other plots in the manuscript. Please harmonize.
We now use consistent symbols with Fig.8 ('O' for in-scattering and 'X' for scattering) and plot all points in black to avoid confusion.

Is there a reason for why the scattered dots to be located on a horizontal line?
"In the scatter plots, the data points are located on horizontal lines because the input cloud fraction and optical thickness values are discretized." This explanation has been added.

Please mention why it is like that. Also, I don't fully understand what the linear regression lines in this figure tend to show/prove. Could you please elaborate on that?
Included the following explanations:
"The slope of the linear regression line is 1.03 and the correlation coefficient is 0.96. These values show that although there is a large spread in the retrieved cloud fraction, the biases by in-scattering and shadowing almost cancel each other in this particular case."
"Generally, we find a good correlation between input and retrieved optical thickness, but there is a significant bias towards too small retrieved optical thicknesses (the regression line is shifted to the left compared to the one-one line)."

13. Line 311 (Section 4.1): lines -> dots.
Done.

14. Lines 331-333 (Section 4.1): this part seems to be about the sun glint only. It would be beneficial to give similar explanations for the cloud bow as well.

The explanations for the cloud bow were given before in lines 326ff:

"In the cloudbow, the intensity I is only weakly influenced by 3D scattering effects, i.e. the solid lines for the broken cloud fields lie on top of the dashed lines showing the IPA calculations. |Q|, which saturates quickly for $\tau \geq 3$, is slightly larger for 3D compared to IPA. This difference is enhanced in the degree of polarization shown in the right panels"

15. Line 350 (Section 5): Is there a reference paper for the EUREC4A campaign? Seems like Stevens et al. 2021 is the one. If so, please include it here.
Moved reference to the right place.

16. Figure 12: maybe worth it to say a few words on the difference among the points that fall exactly on the 1:1 line and those that fall apart from that.
Included the following sentence:
"Only the points corresponding to a cloud fraction of 1 (fully cloudy) lie exactly on the one-one line."

17. Line 418 (Section 5.1): … tau_1 and c_1 are, "respectively", … .
Done.

18. Line 434 (Section 5.1): Do you mean that the "difference" in the cloud optical thickness is small?
No, we mean that the optical thickness in scene 4 is larger than in the previously discussed scenes.

19. Line 452 (Section 5.1): has -> have ; (i.e., smaller than 0.2).
Done.

20. General comment (Section 5): merge section 5 with subsection 5.1. As there is no subsection 5.2, I don't see a need for having 5 and then 5.1.
Yes, we agree. The sections are now merged.

21. Section 6 (Conclusions and Outlook): Outlook -> outlook
Done.

22. Line 459 (Section 6): … optical thickness of liquid clouds over ocean …
Included.

23. Line 475 (Section 6): This demonstrates that …"
Done.